# Seeing is not always believing: Benchmarking Human and Model Perception of AI-Generated Images

**Zeyu Lu**[1,2,*]  **Di Huang**[2,3,*]  **Lei Bai**[2,*,†]  **Jingjing Qu**[2]  **Chengyue Wu**[4]
**Xihui Liu**[4]  **Wanli Ouyang**[2]
[1] Shanghai Jiao Tong University  [2] Shanghai AI Laboratory
[3] The University of Sydney  [4] The University of Hong Kong

## Abstract

Photos serve as a way for humans to record what they experience in their daily lives, and they are often regarded as trustworthy sources of information. However, there is a growing concern that the advancement of artificial intelligence (AI) technology may produce fake photos, which can create confusion and diminish trust in photographs. This study aims to comprehensively evaluate agents for distinguishing state-of-the-art AI-generated visual content. Our study benchmarks both human capability and cutting-edge fake image detection AI algorithms, using a newly collected large-scale fake image dataset **Fake2M**. In our human perception evaluation, titled **HPBench**, we discovered that humans struggle significantly to distinguish real photos from AI-generated ones, with a misclassification rate of **38.7%**. Along with this, we conduct the model capability of AI-Generated images detection evaluation **MPBench** and the top-performing model from MPBench achieves a **13%** failure rate under the same setting used in the human evaluation. We hope that our study can raise awareness of the potential risks of AI-generated images and facilitate further research to prevent the spread of false information. More information can refer to https://github.com/Inf-imagine/Sentry.

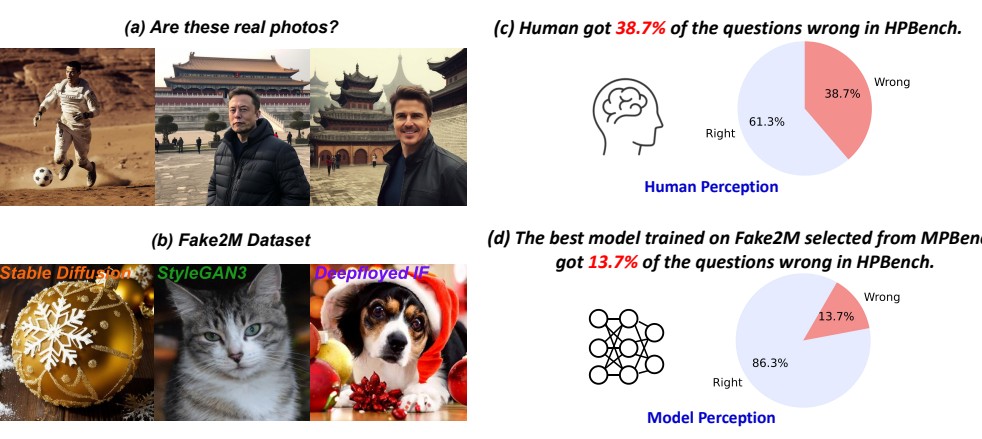

*(a) Are these real photos?*

*(b) Fake2M Dataset*

*(c) Human got 38.7% of the questions wrong in HPBench.*

Wrong

38.7%

Right

61.3%

**Human Perception**

*(d) The best model trained on Fake2M selected from MPBench got 13.7% of the questions wrong in HPBench.*

Wrong

13.7%

Right

86.3%

**Model Perception**

Figure 1: *Left:* (a) Determining whether an image is real is a difficult problem. (b) We introduce a new, large-scale and diverse dataset, named "Fake2M". *Right:* (c) The average human ability to distinguish between high-quality AI-generated images and real images is only 61% in HPBench. (d) The highest performing model trained on the Fake2M dataset, selected from MPBench, achieves an accuracy of 87% on HPBench.

---

[*]Equal contribution.  [†]Corresponding author: bailei@pjlab.org.cn.

37th Conference on Neural Information Processing Systems (NeurIPS 2023) Track on Datasets and Benchmarks.

# 1 Introduction

Photography, which captures images by recording light, has become an integral part of human society, serving as a vital medium for recording real visual information. Its diverse applications range from documenting historical events and scientific discoveries to immortalizing personal memories and artistic expressions. Since the advent of photography with Joseph Nicéphore Niépce's first photograph of photographs in 1826, the global collection of photographs has expanded to an estimated 12.4 trillion photos. This impressive number continues to rise with an annual increase of 1.72 trillion photos (10–14% increasing rate every year) [7].

In contrast to traditional photography, AI-driven image generation harnesses neural networks to learn synthesis rules from extensive image datasets, offering a novel approach for creating high-quality yet fake visual content. By taking image descriptions or random noise as input, AI generates one or several images for users. Early methods employed GANs [12, 24, 29, 31–33, 60] as the generative architecture, while more recent diffusion-based techniques [10, 18, 20, 27, 28, 30, 38, 41, 43, 45–48, 51, 51–54, 61] have showcased improved diversity and generation quality. Therefore, AI-generated contents (AIGC) have gained popularity across various applications, such as AI-assisted design.

Following the rapid advancements in AI-driven image generation algorithms, AI is now capable of producing a wide array of image styles, including those that closely resemble real photographs. At the Sony World Photography Awards in April 2023, the winner Eldagsen refused the award after revealing that his creation is made using AI. Organizers of the award said that Eldagsen had misled them about the extent of AI that would be involved [3]. Consequently, a critical question arises: *Can humans find a reliable solution for distinguishing whether an image is AI-generated?* This inquiry fundamentally questions the reliability of image information for conveying truth. Utilizing generated images to convey false information can have significant social repercussions, often misleading people about nonexistent events or propagating incorrect ideas, as shown in Fig. 1. A recent example involves the circulation of fake images depicting Trump's arrest [2], which garnered substantial social attention.

In order to tackle the challenge of discerning AI-generated images, we carried out an extensive study focusing on human capability and the proficiency of cutting-edge fake image detection AI algorithms. For human capability, we conduct a human evaluation involving participants from diverse backgrounds to determine their ability to distinguish real and fake images. Each participant is tasked with completing a test consisting of 100 randomized questions, where they must decide whether an image is real or generated by AI. As for the evaluation of fake image detection models, existing datasets fell short in supporting this study due to their limited size and inclusion of outdated AI-generated images. Consequently, we assembled a novel dataset, called **Fake2M**, composed of state-of-the-art (SOTA) AI-generated images and real photographs sourced from the internet. Fake2M is a large-scale dataset housing over 2 million AI-generated images from three different models, tailored for training fake image detection algorithms.

For human evaluation of fake image detection task, our key finding underscores that state-of-the-art AI-generated images can indeed significantly deceive human perception. According to our results, participants achieved an average accuracy of only 61.3%, implying a misclassification rate of 38.7%, thus demonstrating the challenge they faced in accurately discerning real images from those generated by AI. Our study also reveals that humans are better at distinguishing AI-generated portrait images compared to other types of AI-generated images.

Turning to the model evaluation for the fake image detection task, our experiments yielded several key insights. Firstly, no single model consistently outperforms the others across all training datasets, suggesting that the optimal model is largely dependent on the specific dataset in use. This underscores the importance of developing a robust model that can consistently achieve superior performance across all training datasets. Secondly, our findings underscored the benefits of diverse training datasets, with models demonstrating improved overall accuracy when exposed to a broader range of visual styles and variations. Lastly, we observed variability in model accuracy when the same model trained on identical datasets was validated using different generation models. This demonstrates that variations in validation dataset generation may influence the performance of model.

In this study, we benchmark the ability of both human and cutting-edge fake image detection AI algorithms. While advancements in image synthesis have enabled individuals to create visually appealing images with quality approaching real photographs, our findings reveal the potential risks associated

with using AIGC for spreading false information and misleading viewers. Our contributions can be listed as follows:

- We introduce a new, large-scale dataset, named "**Fake2M**". To the best of our knowledge, this is the largest and most diverse fake image dataset, designed to stimulate and foster advancements in fake image detection research.
- We establish **HPBench**, a unique benchmark that comprehensively assesses the human capability to discern fake AI-generated images from real ones.
- We establish **MPBench**, an extensive and comprehensive benchmark including 11 fake validation datasets, designed to rigorously assess the model capability for identifying fake images generated by the most advanced generative models currently available.

## 2  Dataset Collection and Generation

### 2.1  Collect Data for Human Evaluation

Table 1: **Number of photographic images used in HPBench across eight categories.**

| Category | Multiperson | Landscape | Man | Woman | Record | Plant | Animal | Object | All |
|---|---|---|---|---|---|---|---|---|---|
| **Number of AI-Generated Images** | 10 | 27 | 17 | 30 | 15 | 13 | 29 | 10 | 151 |
| **Number of Real Images** | 12 | 26 | 44 | 49 | 21 | 18 | 53 | 21 | 244 |

We collect AI-generated images and a set of real photographs across eight categories: Multiperson, Landscape, Man, Woman, Record, Plant, Animal, and Object, as shown Tab. 1.

**Collecting AI-generated photorealistic images.** Firstly, we utilize Midjourney-V5 [6], the state-of-the-art image generation model, to create photorealistic images of the aforementioned eight categories. For each category, we employ diverse prompts to ensure maximum variation. We use specialized prompt suffixes such as "normal color, aesthetic, shocking, very detailed, photographic, 8K, HDR" to improve the authenticity of the images. We notice that in real-world scenarios, people use AI to generate images with the intention of creating high-quality images without visual defects, and users will select the best image from multiple generated images. Therefore, we employ the expertise of annotators to filter out low-quality AI-generated images that can be easily determined as fake photos at the first glance.

**Collecting real photos.** We collect real photos from 500px [1] and Google Images [5] by searching for photos with the same text prompts used for creating AI-generated images in the previous paragraph.

### 2.2  Collect Data for Model Evaluation

Our objective is to investigate whether models can distinguish if an image is AI-generated or not. We constructed 3 training fake datasets with about 2M images, named **Fake2M**, and 11 validation fake datasets with about 257K images using different latest modern generative models, which contain the SOTA Diffusion models (Stable Diffusion [46], IF [4]), the SOTA GAN model (StyleGAN3 [31]), the SOTA autoregressive model (CogView2 [19]), and the SOTA generative model (Midjounrey [6]), as shown in Tab. 2. We describe the details of our datasets in the following subsections.

**Collecting training datasets.** For the text-to-image generation model, we use the first 1M captions from CC3M to generate the corresponding fake images. For the class conditional generation model, we generate the fake images directly. We describe the specific settings of the 3 training fake datasets in Tab 2, as follows: **(1) "SD-V1.5Real-dpms-25":** Stable Diffusion v1.5 Realistic Vision V2.0 [8] (the top popular models in CIVITAI) with DPM-Solver 25 steps to generate the corresponding fake images. **(2) "IF-V1.0-dpms++-25":** IF v1.0 [4] with DPM-Solver++ [37] 25 steps to generate the corresponding fake images. **(3) "StyleGAN3":** To match the training datasets domain in StyleGAN3 [31] and to maximize the diversity of the model, we use StyleGAN3-t-ffhq to generate the 35K fake images, StyleGAN3-r-ffhq to generate the 35K fake images, StyleGAN3-t-metfaces to generate the 650 fake images, StyleGAN3-r-metfaces to generate the 650 fake images, stylegan3-t-afhqv2 to generate the 8K fake images and stylegan3-r-afhqv2 to generate the 8K fake images.

Table 2: **Detailed information of the datasets used in MPBench. R** denotes the dataset consisting entirely of real images. **F** denotes the dataset consisting entirely of fake images. ✔ denotes existing datasets. ✗ denotes the datasets provided in this work. "Diff" refers to diffusion model, "AR" refers to autoregressive model and "Unk." refers to unknown model.

| Dataset | CC3M-Train | StyleGAN3-Train | SD-V1.5Real-dpms-25 | IF-V1.0-dpms++-25 | StyleGAN3 | ImageNet-Test | CelebA-HQ-Train | CC3M-Val | SD-V2.1-dpms-25 | SD-V1.5-dpms-25 | SD-V1.5Real-dpms-25 | IF-V1.0-dpms++-10 | IF-V1.0-dpms++-25 | IF-V1.0-dpms++-50 | IF-V1.0-ddim-50 | IF-V1.0-ddpms-50 | Cogview2 | Midjourney | StyleGan3 |
|---|---|---|---|---|---|---|---|---|---|---|---|---|---|---|---|---|---|---|---|
| Category | Train | | | | | Validate | | | | | | | | | | | | | |
| | R | R | F | F | F | R | R | R | F | F | F | F | F | F | F | F | F | F | F |
| Generator | - | - | Diff. | Diff. | GAN | - | - | - | Diff. | Diff. | Diff. | Diff. | Diff. | Diff. | Diff. | Diff. | AR | Unk. | GAN |
| Numbers | 1M | 87K | 1M | 1M | 87K | 100K | 24K | 15K | 15K | 15K | 15K | 15K | 15K | 15K | 15K | 15K | 22K | 5.5K | 60K |
| This work | ✗ | ✗ | ✔ | ✔ | ✔ | ✗ | ✗ | ✗ | ✔ | ✔ | ✔ | ✔ | ✔ | ✔ | ✔ | ✔ | ✔ | ✔ | ✔ |

In order to align with the domain of former fake datasets, we used the corresponding real images datasets and the specific settings of the 2 training real datasets in Tab 2, as follows: **(1) "CC3M-Train":** To conform with the former fake datasets, we use the corresponding real images from CC3M [50]. **(2) "StyleGAN3-Train":** To match the former fake dataset, we use the StyleGAN3 training datasets: FFHQ [32], AFHQv2 [14] and MetFaces [31].

**Collecting validation datasets.** For the text-to-image generation model, we use the whole 15K captions from CC3M validation dataset to generate the corresponding fake images. For the class conditional generation model, we generate fake images directly. For Midjourney, we crawled 5.5K images from the community as a validation set. We describe the specific settings of the 11 validation fake datasets in Tab 2, as follows: **(1) "SD-V2.1-dpms-25":** Stable Diffusion v2.1 with DPM-Solver [36] 25 steps to generate the corresponding fake images. **(2) "SD-V1.5-dpms-25":** Stable Diffusion v1.5 with DPM-Solver 25 steps to generate the corresponding fake images. **(3) "SD-V1.5Real-dpms-25":** Stable Diffusion v1.5 Realistic Vision V2.0 with DPM-Solver 25 steps to generate the corresponding fake images. **(4) "IF-V1.0-dpms++-10":** IF v1.0 with DPM-Solver++ 10 steps to generate the corresponding fake images. **(5) "IF-V1.0-dpms++-25":** IF v1.0 with DPM-Solver++ 25 steps to generate the corresponding fake images. **(6) "IF-V1.0-dpms++-50":** IF v1.0 with DPM-Solver++ 50 steps to generate the corresponding fake images. **(7) "IF-V1.0-ddim-50":** IF v1.0 with DDIM [51] 50 steps to generate the corresponding fake images. **(8) "IF-V1.0-ddpm-50":** IF v1.0 with DDPM [28] 50 steps to generate the corresponding fake images. **(9) "Cogview2":** Cogview2 to generate the corresponding fake images. **(10) "Midjourney":** We crawl 5K images from the community as a validation set. **(11) "StyleGan3":** To maximize the diversity of the model, we generate 10K images each using StyleGAN3-t-ffhq, StyleGAN3-r-ffhq, StyleGAN3-t-metfaces, StyleGAN3-r-metfaces, stylegan3-t-afhqv2 and stylegan3-r-afhqv2, for a total of 60K images.

We use 3 common real datasets as our validation dataset and the specific settings as follows: **(1) "ImageNet-Test":** The test dataset of ImageNet1k [17]. **(2) "CelebA-HQ-Train":** The training dataset of CelebA-HQ [34]. **(3) "CC3M-Val":** We use the validation dataset of CC3M as our validation dataset.

## 2.3 Comparison with Other Datasets

As shown in Tab. 3, our dataset is the largest and most diverse public general fake image dataset, designed to stimulate and foster advancements in fake image detection research.

Table 3: **Comparison with other fake image datasets.** "Content Type" means the type of the content in each dataset ("Face" means that the content in this dataset is mostly faces, such as FFHQ [32]. "Object" means that the content in this dataset is mainly composed of a limited number of objects, such as ImageNet [17]. "General" means that the content in this dataset is general, not limited to some objects, faces or art, such as CC3M [50]). "Generator Type" means the type of generator used in our dataset. "Public" means the dataset is publicly accessible. "Fake Image Number" represents the number of fake images provided by this dataset.

| Dataset | Content Type | Generator Type | | | Public | Generator Number | Fake Image Number |
| | | GAN | Diffusion | AutoRegressive | | | |
| --- | --- | --- | --- | --- | --- | --- | --- |
| FakeSpotter [56] | Face | ✔ | ✘ | ✘ | ✘ | 7 | 5K |
| DFFD [16] | Face | ✔ | ✘ | ✘ | ✔ | 8 | 240K |
| APFDD [23] | Face | ✔ | ✘ | ✘ | ✘ | 1 | 5K |
| ForgeryNet [26] | Face | ✔ | ✘ | ✘ | ✔ | 15 | 1.4M |
| DeepArt [58] | Art | ✘ | ✔ | ✘ | ✔ | 5 | 73K |
| CNNSpot [57] | Object | ✔ | ✘ | ✘ | ✔ | 11 | 362K |
| IEEE VIP Cup [55] | Object | ✔ | ✔ | ✘ | ✘ | 5 | 7K |
| CiFAKE [11] | Object | ✘ | ✔ | ✘ | ✔ | 1 | 60K |
| **Ours** | **General** | ✔ | ✔ | ✔ | ✔ | **11** | **2.3M** |

# 3 HPBench: Human Perception of AI-Generated Images Evaluation

Our objective is to investigate whether humans can distinguish if an image is AI-generated or not. We collect a set of AI-generated photorealistic images and real photographs, and conduct a human evaluation on the collected images to build **HPBench**. We describe the details of our data collection, human evaluation, and metrics for analyzing the results in the following subsections.

## 3.1 Evaluation Setup

Table 4: **Number of participants across different backgrounds.** "w/ AIGC" refers to participants who have played with AIGC. "w/o AIGC" refers to participants who have not played with AIGC.

| | Gender | | Background | | Age | | All |
| **Category** | Male | Female | w/ AIGC | w/o AIGC | 20~29 | 30~45 | |
| --- | --- | --- | --- | --- | --- | --- | --- |
| **Number** | 31 | 19 | 27 | 23 | 42 | 8 | 50 |

**A high-quality fifty-participant human evaluation.** In order to ensure comprehensiveness, fairness, and quality of the evaluation, we recruit a total of 50 participants to participate in our human evaluation instead of using crowdsourcing and make efforts to ensure the diversity of the participants, as shown in Tab. 4. Each participant is asked to complete a questionnaire consisting of 100 questions to determine whether the image is generated by AI or not without any time limit.

Table 5: **Human evaluation metrics across nine categories using all participant data.**

| | Category | | | | | | | | |
| **Metric** | All | Multiperson | Landscape | Man | Woman | Record | Plant | Animal | Object |
| --- | --- | --- | --- | --- | --- | --- | --- | --- | --- |
| **Accuracy** ↑ | 0.6134 | 0.6750 | 0.5650 | 0.6433 | 0.6637 | 0.6233 | 0.5983 | 0.6133 | 0.5083 |
| **Precision** ↑ | 0.6278 | 0.7075 | 0.5657 | 0.6666 | 0.6765 | 0.6340 | 0.6213 | 0.6156 | 0.5112 |
| **Recall** ↑ | 0.5577 | 0.5966 | 0.5714 | 0.5733 | 0.6275 | 0.5833 | 0.5033 | 0.6033 | 0.3800 |
| **FOR** ↓ | 0.3981 | 0.3487 | 0.4358 | 0.3742 | 0.3473 | 0.3858 | 0.4173 | 0.3888 | 0.4933 |

**Evaluation metrics.** We employ four commonly used evaluation metrics to analyze our results and highlight their respective meanings in the context of our problem. We define positive samples as AI-generated images and negative samples as real images for our problem, and then calculate Accuracy, Precision, Recall, and False Omission Rate (FOR) in the context of our problem in Tab. 5.

### 3.2 Results and Analysis

#### 3.2.1 Overall Ability to Distinguish Real and AI-generated Images

**Results.** The results are shown in Fig. 2. Our study indicates that participants on average are able to correctly distinguish 61.3% of the images, while 38.7% of the images are misclassified. The highest-performing participant is able to correctly distinguish 73% of the images, while the lowest-performing participant is only able to correctly distinguish 40% of the images. These observations demonstrate that a combination of real and AI-generated fake images can easily deceive people. Moreover, the results reveal that humans have an average probability of 66.9% of correctly identifying real photos from the internet, whereas for AI-generated images, people are more likely to be misled and incorrectly identify them as real with an average probability of 44.2%.

**Analysis.** An intriguing observation from the above data is that even for real images collected from the Internet, participants are only able to correctly identify 66.9% of the images that should have been correctly sorted out 100% of the time. This finding demonstrates that AI-generated images not only convey incorrect information to humans but also erode people's trust in accurate

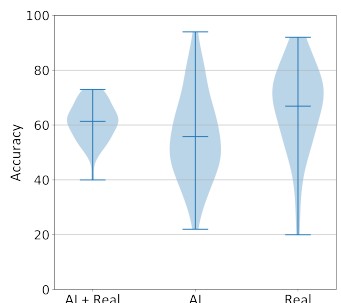

Figure 2: **Human evaluation score distribution using all participant data.** "AI" represents only counting the AI-generated images. "Real" represents only counting the real images. "AI + Real" represents counting all the images.

information. Additionally, our study reveals that humans possess a superior ability to identify real photos than fake photos, which can be attributed to their lifetime experience. Despite this, the difference between real image perception and AI-generated image perception is only 11.1%, and we believe that future generative AI models could further narrow this gap.

#### 3.2.2 Distinguishing Abilities of Participants with Various Personal Backgrounds

**Results.** We explore the effect of AIGC background (refers to participants who have played with AIGC) in our human evaluation. In Fig. 3, we can see that individuals with AIGC background scored slightly better than those without AIGC background (+2.7%). Interestingly, their AIGC background seem to have slight effects on their ability to identify real images (+0.7%). However, when it comes to AI-generated images, participants with AIGC background performs significantly better, with a boost of 3.7%.

**Analysis.** Our research and analysis demonstrate that knowledge and experience with AIGC do not play a significant role in their ability to distinguish between real and AI-generated images. Specifically, when it comes to AI-generated images, participants with AIGC background have a slightly improved ability to distinguish between the two categories. Furthermore, this suggests that additional training and exposure to generative models may be beneficial for individuals by helping them make more informed decisions and avoid any potential risks from fake images.

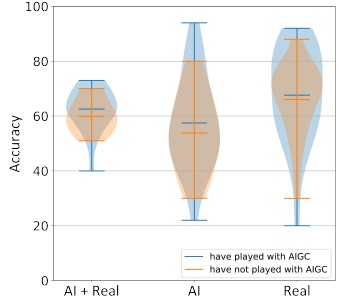

Figure 3: **Human evaluation score distribution calculated by data from participants with AIGC background and without AIGC background.**

#### 3.2.3 Distinguishability of different photo categories

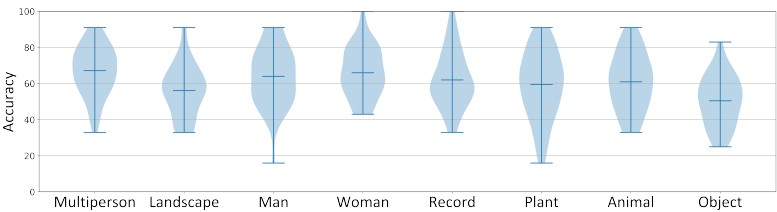

Figure 4: **Human evaluation score distribution across eight categories using all participant data.**

**Results.** We aim to investigate the difficulties for humans to distinguish real and fake images of different image categories. The results are presented in Fig. 4 and Tab. 5. It is observed that the participants had varying ability levels in distinguishing real or fake images from different categories. For instance, the participants can distinguish real or fake images from the category *Multiperson* with a relatively high accuracy rate of 67.5%, whereas they can only correctly distinguish real or fake from the category *Object* with a much lower accuracy rate of 50.8%. Our results indicate a significant 16.7% difference between the accuracy rates of the easiest and most challenging categories. These findings suggest that there may be significant differences in the way humans perceive different photo categories, which could have important implications for further research in this area. It also indicates that the current AI-based generative models may be good at generating some categories but not so good at generating other categories.

**Analysis.** The presented data offers valuable insights into the strengths and limitations of generative AI. The study indicates that current state-of-the-art AI image generation models excel in creating *Object* images that are incredibly realistic. However, AI still struggles when it comes to generating human images. In Tab. 5, *Multiperson*, *Man*, and *Woman* categories are easier to be distinguished as real or fake images compared to others. This phenomenon may be attributed to the fact that the human is more sensitive to images of humans, which is an essential aspect of our cognitive processing.

Table 6: **Statics of judgement criteria from participants correctly identifying fake images.**

| Category | Detail | Smooth | Blur | Color | Shadow & Light | Daub | Rationality | Intuition |
|---|---|---|---|---|---|---|---|---|
| **Number** | 332 | 205 | 142 | 122 | 95 | 59 | 57 | 169 |
| **Percent** | 28% | 17% | 12% | 10% | 8% | 5% | 5% | 14% |

#### 3.2.4 Results of the Judgment Criteria and Analysis of the AIGC Defects

**Results.** We predefined eight judgment criteria options based on our experience and the specific statistics of judgment criteria for correctly selecting the AI-generated images are shown in Tab. 6. The most common issues are "Detail problem" and "Smooth problem" with a high rate of 28% and 17%, respectively. It is also worth noting that the proportion of participants who choose "Intuition" is about 14%, indicating that it is difficult for people to describe obvious defects in AI-generated images, even if they can successfully identify the AI-generated images.

**Analysis.** Our experiment has revealed that even high-quality AI-generated images still exhibit various imperfections and shortcomings. Furthermore, the current SOTA image generation model has limited capability in generating fine details and often generates portraits that are overly smoothed.

## 4 MPBench: Model Perception of AI-Generated Images Evaluation

Our objective is to investigate whether AI models can distinguish if an image is AI-generated or not. We conduct a large and comprehensive model evaluation using **Fake2M** training dataset and 11 fake validation datasets to build **MPBench**.

### 4.1 Experiments Setup

We compare the following SOTA methods: (1) Wang *et al.* [57] proposed to finetune a classification network to give a real/fake decision for an image using *Blur* and *JPEG* augmentation with a pre-trained visual backbone, such as ResNet-50 [25] and ConvNext-S [35]. (2) Ojha *et al.* [42] proposed to use a frozen CLIP-ViT-L [44] for backbone and train the last linear layer for the same task [42].

To ensure a fair comparison, we train each model using four different dataset settings, consisting of 1 million fake images and an equivalent number of real images: (1) Dataset Setting A: "SD-V1.5Real-dpms-25" dataset with 1M fake images and "CC3M-Train" dataset with 1M real images. (2) Dataset Setting B: "IF-V1.0-dpms++-25" dataset with 1M fake images and "CC3M-Train" dataset with 1M real images. (3) Dataset Setting C: "StyleGAN3" dataset with 87K fake images and "StyleGAN3-Train" with 87K real images. (4) Dataset Setting D: "SD-V1.5Real-dpms-25, IF-V1.0-dpms++-25, StyleGAN3, CC3M-Train, StyleGAN3-Train" dataset with 460K fake images selected from "SD-V1.5Real-dpms-25" dataset, 460K fake images selected from "IF-V1.0-dpms++-25" dataset, 87K

Table 7: **Quantitative comparison of five models under four training dataset settings with fourteen validation datasets.** "Diff" refers to diffusion model, "AR" refers to autoregressive model and "Unk." refers to unknown model. **Real (R)** denotes the dataset consisting entirely of real images. **Fake (F)** denotes the dataset consisting entirely of fake images.

| Model | Training Dataset | ImageNet-Test | CelebA-HQ-Train | CC3M-Val | Average Acc. | SD-V2.1-dpm-25 | SD-V1.5-dpm-25 | SD-V1.5Real-dpm-25 | IF-V1.0-dpm++-10 | IF-V1.0-dpm++-25 | IF-V1.0-dpm++-50 | IF-V1.0-ddim-50 | IF-V1.0-ddpm-50 | Cogview2 | Midjourney | StyleGAN3 | Average Acc. | Total Average Acc. |
|---|---|---|---|---|---|---|---|---|---|---|---|---|---|---|---|---|---|---|
| | | | | **Real** | | | | | | **Fake** | | | | | | | | **R+F** |
| | | - | - | - | - | Diff. | Diff. | Diff. | Diff. | Diff. | Diff. | Diff. | Diff. | AR | Unk. | GAN | - | - |
| ConvNext-S(B+J 0.1) | | 95.3 | 99.9 | 99.9 | 98.3 | 48.6 | 99.9 | 85.1 | 92.9 | 57.6 | 55.9 | 41.1 | 72.6 | 55.7 | 41.6 | 37.4 | 62.5 | 70.2 |
| ConvNext-S(B+J 0.5) | Dataset Setting A: | 95.9 | 99.9 | 99.9 | _98.5_ | 53.5 | 100 | 83.3 | 91.1 | 50.2 | 49.6 | 35.3 | 66.9 | 54.9 | 44.7 | 35.7 | 60.4 | 68.6 |
| ResNet50(B+J 0.1) | SD-V1.5Real-dpms-25 (1M) | 93.0 | 95.3 | 95.6 | 94.6 | 71.7 | 71.8 | 98.6 | 57.2 | 26.6 | 29.0 | 23.6 | 47.9 | 11.8 | 40.3 | 9.1 | 44.3 | 55.1 |
| ResNet50(B+J 0.5) | CC3M-Train (1M) | 93.2 | 95.5 | 95.8 | 94.8 | 70.6 | 70.4 | 98.5 | 52.1 | 23.6 | 26.1 | 21.6 | 46.2 | 10.6 | 36.6 | 7.2 | 42.1 | 53.4 |
| CLIP-ViT-L(LC) | | 49.6 | 87.0 | 75.4 | 70.6 | 73.3 | 86.6 | 97.6 | 93.9 | 77.8 | 71.4 | 84.1 | 90.9 | 86.6 | 88.7 | 86.1 | **85.1** | 82.0 |
| ConvNext-S(B+J 0.1) | | 87.6 | 99.9 | 99.9 | _95.8_ | 2.2 | 34.5 | 2.5 | 99.7 | 99.9 | 99.9 | 11.1 | 66.4 | 19.2 | 8.9 | 10.1 | 41.3 | 52.9 |
| ConvNext-S(B+J 0.5) | Dataset Setting B: | 87.8 | 99.9 | 99.9 | _95.8_ | 3.9 | 39.1 | 3.9 | 99.6 | 99.9 | 99.8 | 18.5 | 79.2 | 25.8 | 8.1 | 8.0 | 44.1 | 55.2 |
| ResNet50(B+J 0.1) | IF-V1.0-dpms++-25 (1M) | 89.4 | 95.8 | 95.0 | 93.4 | 37.5 | 56.5 | 20.0 | 84.0 | 95.6 | 91.7 | 39.7 | 69.4 | 45.3 | 15.7 | 8.8 | 51.2 | 60.3 |
| ResNet50(B+J 0.5) | CC3M-Train (1M) | 90.8 | 95.6 | 94.5 | 93.6 | 41.6 | 58.8 | 21.7 | 82.3 | 95.2 | 91.3 | 47.0 | 79.7 | 56.1 | 18.3 | 6.8 | _54.4_ | _62.8_ |
| CLIP-ViT-L(LC) | | 81.5 | 83.3 | 93.0 | 85.9 | 38.2 | 18.5 | 13.1 | 80.4 | 79.7 | 70.9 | 61.1 | 77.7 | 76.6 | 33.7 | 32.8 | 52.9 | 60.0 |
| ConvNext-S(B+J 0.1) | | 58.7 | 81.0 | 62.7 | 67.4 | 44.9 | 42.7 | 39.4 | 40.7 | 42.3 | 42.2 | 35.7 | 39.9 | 53.2 | 41.6 | 69.9 | 44.7 | 49.6 |
| ConvNext-S(B+J 0.5) | Dataset Setting C: | 67.7 | 92.6 | 71.2 | _77.1_ | 33.1 | 33.7 | 32.1 | 33.6 | 36.5 | 34.2 | 35.0 | 35.5 | 44.8 | 28.5 | 42.6 | 35.4 | 44.3 |
| ResNet50(B+J 0.1) | StyleGAN3 (87K) | 31.5 | 14.3 | 39.1 | 28.3 | 67.7 | 62.7 | 68.1 | 65.3 | 71.0 | 67.0 | 58.8 | 60.9 | 60.9 | 81.9 | 90.9 | 68.6 | 60.0 |
| ResNet50(B+J 0.5) | StyleGAN3-Train (87K) | 70.6 | 29.5 | 63.4 | 54.5 | 37.3 | 34.9 | 40.0 | 43.3 | 43.9 | 41.9 | 34.8 | 38.4 | 47.8 | 58.1 | 81.3 | 45.6 | 47.5 |
| CLIP-ViT-L(LC) | | 37.6 | 85.9 | 70.2 | 64.5 | 85.7 | 92.9 | 94.8 | 95.3 | 89.5 | 84.0 | 87.5 | 93.0 | 84.6 | 81.4 | 61.9 | **86.4** | 81.7 |
| ConvNext-S(B+J 0.1) | Dataset Setting D: | 95.4 | 99.9 | 99.9 | 98.4 | 39.9 | 95.1 | 99.9 | 99.7 | 99.8 | 99.7 | 43.1 | 90.4 | 48.3 | 32.4 | 99.8 | 77.1 | 81.6 |
| ConvNext-S(B+J 0.5) | SD-V1.5Real-dpms-25 (460K) | 97.7 | 99.9 | 99.9 | **99.1** | 52.8 | 92.9 | 99.9 | 99.5 | 99.7 | 99.3 | 46.4 | 91.7 | 48.6 | 35.0 | 99.9 | 78.7 | **83.0** |
| ResNet50(B+J 0.1) | IF-V1.0-dpms++-25 (460K) | 77.3 | 93.6 | 91.7 | 87.5 | 83.9 | 90.4 | 96.8 | 91.2 | 90.6 | 86.7 | 52.3 | 82.2 | 56.4 | 49.4 | 80.8 | 78.2 | 80.2 |
| ResNet50(B+J 0.5) | StyleGAN3 (87K) CC3M-Train (1M) | 84.5 | 94.7 | 92.1 | 90.4 | 84.4 | 89.4 | 96.1 | 88.9 | 89.4 | 85.6 | 52.4 | 83.8 | 55.4 | 45.8 | 69.5 | 76.4 | 79.4 |
| CLIP-ViT-L(LC) | StyleGAN3-Train (87K) | 50.4 | 96.0 | 79.8 | 75.4 | 66.9 | 87.8 | 93.6 | 96.1 | 85.9 | 78.9 | 84.9 | 92.6 | 87.4 | 79.8 | 62.8 | _83.3_ | 81.6 |

fake images from "StyleGAN3" dataset, 1M real images from "CC3M-Train" dataset and 87K real images from "StyleGAN3-Train" dataset.

We eval all the models using 11 validation datasets to build **MPBench**, as shown in Tab 7. To clearly show the performance of the models under different validation datasets, we directly report the classification accuracy of each model.

## 4.2 Results and Analysis

### 4.2.1 Comparative Analysis of Accuracy Across Various Models

**Results.** We conducted a systematic analysis of the mean accuracy achieved by different models, each trained using an identical dataset. Our findings are summarized in Tab. 7. Notably, the leading performing models for fake image detection vary depending on the training dataset. For example, under Dataset Setting A, CLIP-ViT-L(LC) excels, whereas under Dataset Setting B, ResNet50 (B+J 0.5) outperforms others. Furthermore, CLIP-ViT-L(LC) achieves the highest fake image detection accuracies of 86.4% and 83.3% under Dataset Setting C and Dataset Setting D respectively. A compelling finding from our study is the absence of a single model that consistently delivers superior performance across all dataset settings for fake image detection. However, our experiments also suggest that ConvNext models are more adept at achieving higher average accuracies in detecting real images. In all four dataset settings, ConvNext consistently achieves top accuracies—98.5%, 95.8%, 77.1%, and 99.1% for Dataset Setting A,B,C,D, respectively.

**Analysis.** Our observations highlight the need for models that perform optimally on both real and fake images. According to our study, while CLIP-ViT-L(LC) most often produces the best results for fake images, ConvNext outperforms CLIP-ViT-L(LC) in real image detection. However, real-world applications necessitate discerning between real and fake images. This underscores the need for future research to strike a balance between detection capabilities for both categories.

### 4.2.2 Comparative Analysis of Accuracy Across Various Training Datasets

**Results.** We conduct an empirical analysis of the average accuracy of various models, each trained using different dataset configurations. The results, as presented in Tab. 7, indicate that Dataset Setting D generally yields superior model performance. ConvNext and ResNet50 models specifically show marked improvements when trained on Dataset Setting D. Meanwhile, CLIP-ViT-L (LC) demonstrates comparable performance across different dataset configurations, attaining accuracies of 82.0% with Dataset Setting A and 81.6% with Dataset Setting D.

**Analysis.** Our experimental findings suggest that the choice of training dataset significantly influences model performance in fake image detection tasks. Notably, a diversified dataset like Dataset Setting D, which includes five distinct generative models, seems to enhance overall model accuracy. This improvement is likely due to the exposure to a broader spectrum of generative styles and variations that a diversified dataset offers. Additionally, our results highlight the aptitude of the proposed dataset, **Fake2M**, which features diverse data sources, for more generalized fake image detection.

### 4.2.3 Comparative Analysis of Accuracy Across Various Validation Datasets

**Results.** In our experiment, we assessed the accuracy of a trained model across various validation datasets. As depicted in Tab. 7, the performance of a model varies significantly based on the generation models, sampling methods, and sampling steps used in the validation set. For instance, the ConvNext-S (B+J 0.5) model, when trained using Dataset Setting D, displayed a broad range of validation results for fake image detection, with accuracies spanning between 35.0% and 99.9%. These findings underscore the influence of validation set characteristics on a model's performance in the task of fake image detection.

**Analysis.** In realistic applications, fake images can originate from a variety of generation models with a diverse range of hyperparameters. Consequently, an ideal fake image detection model should demonstrate consistent proficiency across this broad spectrum of generation settings. However, our experimental results show that existing models do not meet this requirement, indicating a need for the development of new approaches to handle the challenge of detecting fake images generated under varying settings.

### 4.2.4 Evaluate the best model under the same setting used in HPBench.

Based on its highest total average accuracy in MPBench, we select ConvNext-S (B+J 0.5) with Dataset Setting D as the best model. We then evaluate this model under the same setting in HPBench which consists of 50 real images and 50 fake images for testing and it achieves a 13% failure rate.

## 5  Related Work

**Image Generation Models.** State-of-the-art text-to-image synthesis approaches such as DALL·E 2 [45], Imagen [48], Stable Diffusion [46], IF [4], and Midjourney [6] have demonstrated the possibility of that generating high-quality, photorealistic images with diffusion-based generative models trained on large datasets [49]. Those models have surpassed previous GAN-based models [12, 24, 32, 60] in both fidelity and diversity of generated images, without the instability and mode collapse issues that GANs are prone to. In addition to diffusion models, other autoregressive models such as Make-A-Scene [22], CogView [19], and Parti [59] have also achieved amazing performance.

**Fake Images Generation and Detection.** In recent years, there have been many works [9, 13, 15, 21, 39, 40, 42, 57, 62] exploring how to distinguish whether an image is AI-generated. These works focus on fake contents generated by GANs or small generation models [12, 24, 32, 60]. Due to the limited quality of images generated by those methods, it is easy for humans to distinguish whether a photo is AI-generated or not. However, as the quality of generated images continues to improve with the advancement of recent generative models [6, 45, 46, 48], it has become increasingly difficult for humans to identify whether an image is generated by AI or not. The need for detecting fake images has existed even before we had powerful image generators.

# 6 Conclusion

In this study, we present a comprehensive evaluation of both human discernment and contemporary AI algorithms in detecting fake images. Our findings reveal that humans can be significantly deceived by current cutting-edge image generation models. In contrast, AI fake image detection algorithms demonstrate a superior ability to distinguish authentic images from fakes. Despite this, our research highlights that existing AI algorithms, with a considerable misclassification rate of 13%, still face significant challenges. We anticipate that our proposed dataset, **Fake2M**, and our dual benchmarks, **HPBench** and **MPBench**, will invigorate further research in this area and assist researchers in crafting secure and reliable AI-generated content systems. As we advance in this technological era, it is crucial to prioritize responsible creation and application of generative AI to ensure its benefits are harnessed positively for society.

**Acknowledgments and Disclosure of Funding.** This research is funded by Shanghai AI Laboratory. This work is partially supported by the National Key R&D Program of China (NO.2022ZD0160100), (NO.2022ZD0160101). We would like to thank many colleagues for useful discussions, suggestions, feedback, and advice, including: Peng Ye, Min Shi, Weiyun Wang, Yan Teng, Qianyu Guo, Kexin Huang, Yongqi Wang and Tianning Zhang.

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
