# Seeing is not always believing: Benchmarking Human and Model Perception of AI-Generated Images

## A Quick Test: Can you identify which ones are AI-generated images?

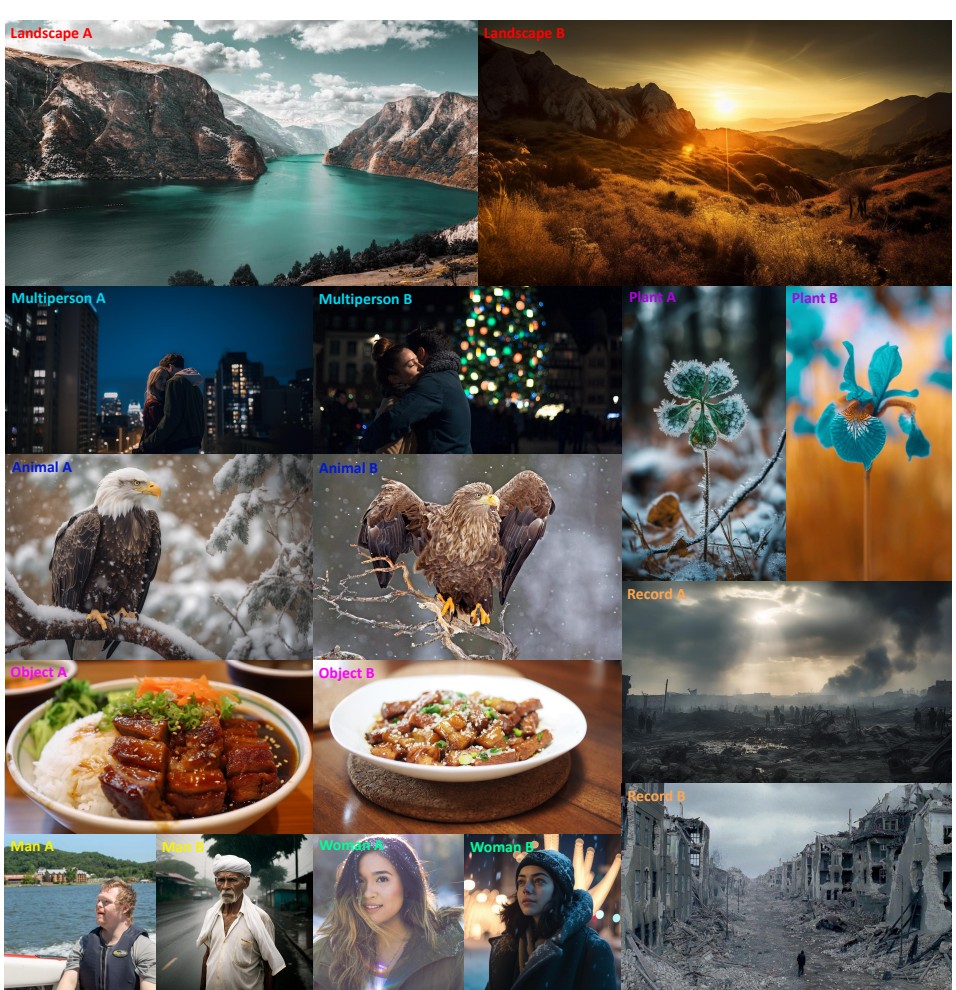

Figure 1: **A quick test:** *Can you identify which ones are AI-generated images?*

**Answer of the Quick Test** The AI-generated images of the Fig. 1 are "Landscape B", "Multiperson A", "Plant A", "Animal A", "Record A", "Object A", "Man B", "Woman B", respectively.

37th Conference on Neural Information Processing Systems (NeurIPS 2023) Track on Datasets and Benchmarks.

Table 1: **Detailed information of the datasets used in MPBench. R** denotes the dataset consisting entirely of real images. **F** denotes the dataset consisting entirely of fake images. ✔ denotes existing datasets. ✗ denotes the datasets provided in this work. "Diff" refers to diffusion model, "AR" refers to autoregressive model and "Unk." refers to unknown model. "Resolution" refers to the resolution of the fake images in the dataset. "Caption" refers to the caption used in text-to-image generation models to generate the corresponding dataset.

| Dataset | CC3M-Train | StyleGAN3-Train | SD-V1.5Real-dpms-25 | IF-V1.0-dpms++25 | StyleGAN3 | ImageNet-Test | CelebA-HQ-Train | CC3M-Val | SD-V2.1-dpms-25 | SD-V1.5-dpms-25 | SD-V1.5Real-dpms-25 | IF-V1.0-dpms++-10 | IF-V1.0-dpms++-25 | IF-V1.0-dpms++-50 | IF-V1.0-ddim-50 | IF-V1.0-dpms-50 | Cogview2 | Midjourney | StyleGan3 |
|---|---|---|---|---|---|---|---|---|---|---|---|---|---|---|---|---|---|---|---|
| Category | Train | | | | | Validate | | | | | | | | | | | | | |
| | R | R | F | F | F | R | R | R | F | F | F | F | F | F | F | F | F | F | F |
| Generator | - | - | Diff. | Diff. | GAN | - | - | - | Diff. | Diff. | Diff. | Diff. | Diff. | Diff. | Diff. | Diff. | AR | Unk. | GAN |
| Numbers | 1M | 87K | 1M | 1M | 87K | 100K | 24K | 15K | 15K | 15K | 15K | 15K | 15K | 15K | 15K | 15K | 22K | 5.5K | 60K |
| Resolution | - | - | 512 | 256 | (>=)512 | - | - | - | 512 | 512 | 512 | 256 | 256 | 256 | 256 | 256 | 480 | (>=)640 | (>=)512 |
| Caption | - | - | CC3M-train(first 1M) | CC3M-train(first 1M) | - | - | - | - | CC3M-val | CC3M-val | CC3M-val | CC3M-val | CC3M-val | CC3M-val | CC3M-val | CC3M-val | - | - | - |
| Seed | - | - | 420 | 420 | - | - | - | - | 420 | 420 | 420 | 420 | 420 | 420 | 420 | 420 | - | - | - |
| CFG-Scale | - | - | 7 | 7 | - | - | - | - | 7 | 7 | 7 | 7 | 7 | 7 | 7 | 7 | - | - | - |
| This work | ✗ | ✗ | ✔ | ✔ | ✔ | ✗ | ✗ | ✗ | ✔ | ✔ | ✔ | ✔ | ✔ | ✔ | ✔ | ✔ | ✔ | ✔ | ✔ |

Table 2: **Detailed information of the diffusion datasets used in MPBench.**

| Category | Train | | Test | | | | | | | | |
|---|---|---|---|---|---|---|---|---|---|---|---|
| Generators | SD-V1.5Real-dpms-25 | IF-V1.0-dpms++-25 | SD-V2.1-dpms-25 | SD-V1.5-dpms-25 | SD-V1.5Real-dpms-25 | IF-V1.0-dpms++-10 | IF-V1.0-dpms++-25 | IF-V1.0-dpms++-50 | IF-V1.0-ddim-50 | IF-V1.0-dpms-50 |
| Total Numbers | 1M | 1M | 15K | 15K | 15K | 15K | 15K | 15K | 15K | 15K |
| Sampling Steps | 25 | 25 | 25 | 25 | 25 | 10 | 25 | 50 | 50 | 50 |
| Sampling Methods | dpm-sovler [31] | dpm-sovler++ [32] | dpm-sovler [31] | dpm-sovler [31] | dpm-sovler [31] | dpm-sovler++ [32] | dpm-sovler++ [32] | dpm-sovler++ [32] | ddim [49] | ddpm [23]s |
| Seed | 420 | 420 | 420 | 420 | 420 | 420 | 420 | 420 | 420 | 420 |
| CFG-Scale | 7 | 7 | 7 | 7 | 7 | 7 | 7 | 7 | 7 | 7 |
| Model | Stable Diffusion v1.5 Realistic Version | IF v1.0 | Stable Diffusion v2.1 | Stable Diffusion v1.5 | Stable Diffusion v1.5 Realistic Version | IF v1.0 | IF v1.0 | IF v1.0 | IF v1.0 | IF v1.0 |

Table 3: **Detailed information of the StyleGAN3 datasets used in MPBench.**

| Category | Train | | | | | | Validate | | | | | |
|---|---|---|---|---|---|---|---|---|---|---|---|---|
| Generators | stylegan3-r-ffhqu-1024x1024 | stylegan3-t-ffhqu-1024x1024 | stylegan3-r-afhqv2-512x512 | stylegan3-t-afhqv2-512x512 | stylegan3-r-metfaces-1024x1024 | stylegan3-t-metfaces-1024x1024 | stylegan3-r-ffhqu-1024x1024 | stylegan3-t-ffhqu-1024x1024 | stylegan3-r-afhqv2-512x512 | stylegan3-t-afhqv2-512x512 | stylegan3-r-metfaces-1024x1024 | stylegan3-t-metfaces-1024x1024 |
| Total Numbers | 87K | | | | | | 60K | | | | | |
| Numbers | 35K | 35K | 8K | 8K | 0.65K | 0.65K | 10K | 10K | 10K | 10K | 10K | 10K |
| Seeds | 10001~45000 | 10001~45000 | 10001~18000 | 10001~18000 | 10001~10800 | 10001~10800 | 1-10000 | 1-10000 | 1-10000 | 1-10000 | 1-10000 | 1-10000 |
| Matched Dataset | FFHQ (70K) [27] | | AFHQv2 (16K) [12] | | Metfaces (1.3K) [26] | | None | | | | | |

# B Dataset

## B.1 Dataset Configuration for Model Evaluation

We detailed the collection process of our datasets in Section 2.2 of the main paper, now the following will provide more detailed configuration information for each dataset.

We use the default Github repository code of each model to generate our datasets. Detailed information about the training and validation datasets are shown in Tab. 1. We further provide the captions and resolutions used in each specific dataset. For diffusion generation, we use the fixed seed and cfg-scale to generate our datasets. We also use different sampling methods and steps for generation. The detailed information about sampling methods and steps for different diffusion models can be found in Tab. 2. For StyleGAN3 generation, we use 2 models (stylegan3-r-ffhqu-1024x1024, stylegan3-t-ffhqu-1024x1024) to generate 70K face images for training to match the number of FFHQ and 10K face images for testing. We use 2 models (stylegan3-r-afhqv2-512x512, stylegan3-t-afhqv2-512x512) to generate 16K animal faces to match the number of AFHQ-v2 and 10K animal faces for testing. We use 2 models (stylegan3-r-metfaces-1024x1024, stylegan3-t-metfaces-1024x1024) to generate 1.3K art human faces for training to match the number of MetFaces Dataset and 10K art human faces for testing. The detailed information about our StyleGAN3 generation can be found in Tab. 3.

## B.2 Data Content Component Analysis of Training and Validation Dataset

We will analyze the composition of the training and validation dataset in the following two parts and discuss the issue of data imbalance. We also provide a detailed table showing the composition and proportion of different datasets, as shown in Tab. 4.

**Training Dataset.**

• **Fake2M** is composed of 1M fake images generated by the first 1M caption in CC3M using SD-V1.5Real-dpms-25 [44], 1M fake images generated by the first 1M caption in CC3M using IF-V1.0-dpms++-25 [3] and 87K fake images generated using StyleGAN3 [26], as shown in Tab. 1 and Tab 2.

In Fake2M, the number of face data is only 82K, accounting for %4 of the total data 2M, as shown in Tab. 4. There is no content imbalanced problem in Fake2M.

• **Training Dataset Setting A** is composed of 1M fake images generated by the first 1M caption in CC3M using SD-V1.5Real-dpms-25 in Fake2M and the first 1M real images in CC3M.

In Training Dataset Setting A, most of the content is general content. There is no content imbalanced problem in Training Dataset Setting A.

• **Training Dataset Setting B** is composed of 1M fake images generated by the first 1M caption in CC3M using IF-V1.0-dpms++-25 in Fake2M and the first 1M real images in CC3M.

In Training Dataset Setting B, most of the content is general content. There is no content imbalanced problem in Training Dataset Setting B.

• **Training Dataset Setting C** is composed of 87K fake images generated by StyleGAN3 in Fake2M (the detailed content in this dataset can be found in Tab. 3) and the first 1M real images in CC3M.

In training dataset setting C, most of the content is face. There is content imbalanced problem in training dataset setting C. This inclusion was intentional, aiming to specifically investigate the performance implications of face fake images produced by StyleGAN3.

• **Training Dataset Setting D** is composed of 460K fake images generated by the first 460K caption in CC3M using IF-V1.0-dpms++-25 in Fake2M, 460K fake images generated by the first 460K caption in CC3M using SD-V1.5Real-dpms-25 in Fake2M, 87K fake images generated by StyleGAN3, the first 1M real images in CC3M and 87K real images in StyleGAN3 training dataset.

In training dataset setting D, most of the content is general content. The number of fake face data is 82K and real face data is also 82K, accounting for %8 of the total data 2M. There is no content imbalanced problem in training dataset setting D.

**Validation Dataset (MPBench).** In MPBench, most of the content is general content. The number of fake face data is 60K, accounting for %15.3 of the total data 391.5K. The number of real face data is 24K, accounting for %6.1 of the total data 391.5K. There is no content imbalanced problem in training dataset setting D.

From the perspective of the ratio between real and fake images, we observe that the proportion of real and fake images is essentially the same across the four dataset settings and MPBench, as shown in Tab. 4. Therefore, there is no imbalance issue between the number of fake and real images.

## B.3 Quality Analysis

We conducted further analysis of our dataset quality score distributions, as shown in Fig. 2 and Tab. 5. We observed that the majority of our sub dataset have an average score above 0.6 (with Midjourneyv5-5K having an average score of 0.66) and the average score of all images in the dataset is 0.6. These demonstrate that our dataset is a high-quality dataset with a large amount of high-quality images. Only a few datasets (cogview2-22K, IF-ddim-25-15K-1024x1024, IF-ddim-50-15K-1024x1024, stylegan3-r-ffhqu-1024x1024, and stylegan3-r-metfaces-1024x1024) have an average score below 0.6. The distribution of quality scores across the entire dataset demonstrates a balanced mixture of high-quality and low-quality images, as shown in the "all-images" violin plot of Fig. 2. This aligns with our original intention: a fake image detection dataset should encompass both high-quality and low-quality image data. In order to better showcase our dataset, we provided more visualizations about the high quality, mid quality and low quality images in our dataset, as shown in Fig. 3.

Table 4: **Data Content Component Analysis.** "Content" means the type of the content in each dataset ("Face" means that the content in this dataset is mostly faces, such as FFHQ [27]. "Object" means that the content in this dataset is mainly composed of a limited number of objects, such as ImageNet [14]. "General" means that the content in this dataset is general, not limited to some objects, faces or art, such as CC3M [47]). "Each Dataset / Total Number (%)" means the number of images in this dataset and the percentage it contributes to the entire dataset setting setting. "Fake / Total Number (%)" means the number of fake images in the whole dataset setting and the percentage it contributes to the entire dataset setting. "Real / Total Number (%)" means the number of real images in the whole dataset setting and the percentage it contributes to the entire dataset setting.

| Dataset | Fake | | | | Real | | | |
|---|---|---|---|---|---|---|---|---|
| | Name | Content | Each Dataset / Total Number (%) | Fake / Total Number (%) | Name | Content | Each Dataset / Total Number (%) | Real / Total Number (%) |
| Fake2M Dataset | SD-V1.5Real-dpms-25 | General | 1M (47.9%) | | | | | |
| | IF-V1.0-dpms++-25 | General | 1M (47.9%) | 2.08M (100%) | | | | |
| | StyleGAN3 | Face | 87K (4.2%) | | | | | |
| Training Dataest Setting A | SD-V1.5Real-dpms-25 | General | 1M (50%) | 1M (50%) | CC3M-Train | General | 1M (50%) | 1M (50%) |
| Training Dataest Setting B | IF-V1.0-dpms++-25 | General | 1M (50%) | 1M (50%) | CC3M-Train | General | 1M (50%) | 1M (50%) |
| Training Dataest Setting C | StyleGAN3 | Face | 87K (50%) | 87K (50%) | CC3M-Train | General | 87K (50%) | 87K (50%) |
| Training Dataest Setting D | SD-V1.5Real-dpms-25 | General | 460K (21.2%) | | CC3M-Train | General | 1M (46%) | |
| | IF-V1.0-dpms++-25 | General | 460K (21.2%) | 1.08M (50%) | StyleGAN3-Train | Face | 87K (4%) | 1.08M (50%) |
| | StyleGAN3 | Face | 87K (4%) | | | | | |
| Validation Dataset (MPBench) | SD-V2.1-dpm-25 | General | 15K (3.8%) | | ImageNet-Test | Object | 100K (25.5%) | |
| | SD-V1.5-dpm-25 | General | 15K (3.8%) | | CelebA-HQ-Train | Face | 24K (6.1%) | |
| | SD-V1.5Real-dpm-25 | General | 15K (3.8%) | | CC3M-Val | General | 15K (3.8%) | |
| | IF-V1.0-dpm++-10 | General | 15K (3.8%) | | | | | |
| | IF-V1.0-dpm++-25 | General | 15K (3.8%) | | | | | |
| | IF-V1.0-dpm++-50 | General | 15K (3.8%) | 252.5K (64.5%) | | | | 139K (35.5%) |
| | IF-V1.0-ddim-50 | General | 15K (3.8%) | | | | | |
| | IF-V1.0-ddpm-50 | General | 15K (3.8%) | | | | | |
| | Cogview2 | General | 22K (5.6%) | | | | | |
| | Midjourney | General | 5.5K (1.4%) | | | | | |
| | StyleGAN3 | Face | 60K (15.3%) | | | | | |

Table 5: **Quality score distribution statistical information of the dataset.** "all-images" means the quality score distribution of all the images in the dataset. "Mean Score" means the average score of the quality score in the sub dataset. "Min Score" means the minimum score of the quality score in the sub dataset. "Max Score" means the maximum score of the quality score in the sub dataset.

| Sub Dataset | Mean Score | Min Score | Max Score |
|---|---|---|---|
| cogview2-22K | 0.43 | 0.08 | 0.87 |
| IF-ddim-25-15K-1024x1024 | 0.54 | 0.08 | 0.93 |
| IF-ddim-50-15K-1024x1024 | 0.56 | 0.13 | 0.91 |
| IF-ddpm-50-15K-1024x1024 | 0.60 | 0.10 | 0.92 |
| IF-dpmsolver++-10-15K-1024x1024 | 0.63 | 0.12 | **0.95** |
| IF-dpmsolver++-25-15K-1024x1024 | 0.68 | 0.14 | **0.95** |
| Midjourneyv5-5K | 0.66 | 0.15 | 0.91 |
| SDv15-dpmsolver-25-15K | 0.64 | 0.16 | 0.93 |
| SDv15R-dpmsolver-25-15K | 0.64 | 0.20 | 0.93 |
| SDv21-CC15K | **0.68** | 0.19 | 0.92 |
| stylegan3-r-afhqv2-512x512 | 0.65 | 0.21 | 0.89 |
| stylegan3-r-ffhqu-1024x1024 | 0.59 | 0.27 | 0.81 |
| stylegan3-r-metfaces-1024x1024 | 0.53 | 0.19 | 0.78 |
| stylegan3-t-afhqv2-512x512 | 0.66 | 0.25 | 0.89 |
| stylegan3-t-ffhqu-1024x1024 | 0.63 | **0.30** | 0.85 |
| stylegan3-t-metfaces-1024x1024 | 0.55 | 0.23 | 0.79 |
| all-images | 0.60 | 0.08 | **0.95** |

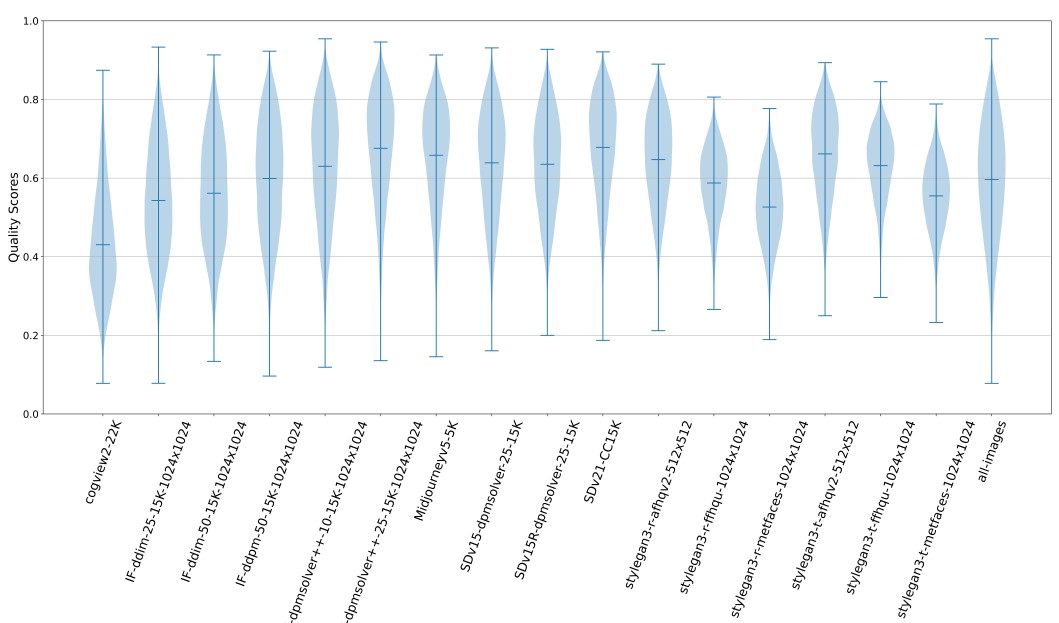

Figure 2: **Quality score distribution of the dataset.** "all-images" means the quality score distribution of all the images in the dataset.

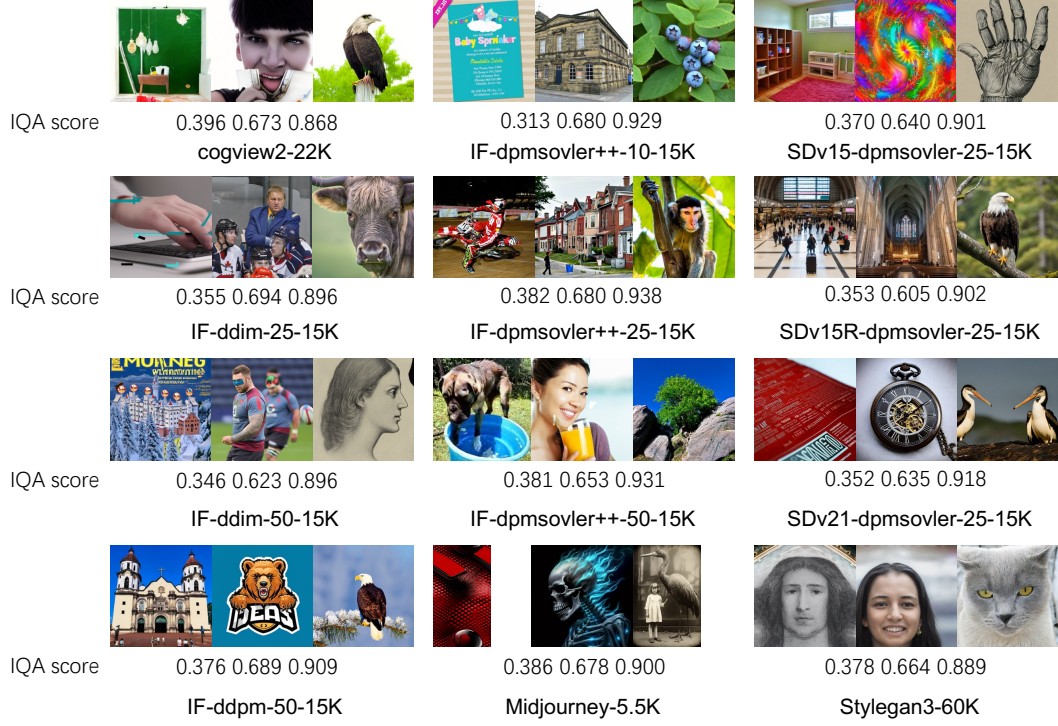

Figure 3: **Image visualization with image quality score.**

# C HPBench

## C.1 Procedures for HPBench

Fig. 4 shows that our HPBench could be divided into three parts. In the first part, we collect realistic AI-generated images and real images across eight categories using the expertise of an annotator to

filter out low-quality AI-generated images. In the second part, we recruit a total of 50 volunteers for human evaluation. For each volunteer, he/she should complete a 100-question questionnaire in our prepared environment. In the third part, We disposal and analyze human evaluation data to draw conclusions.

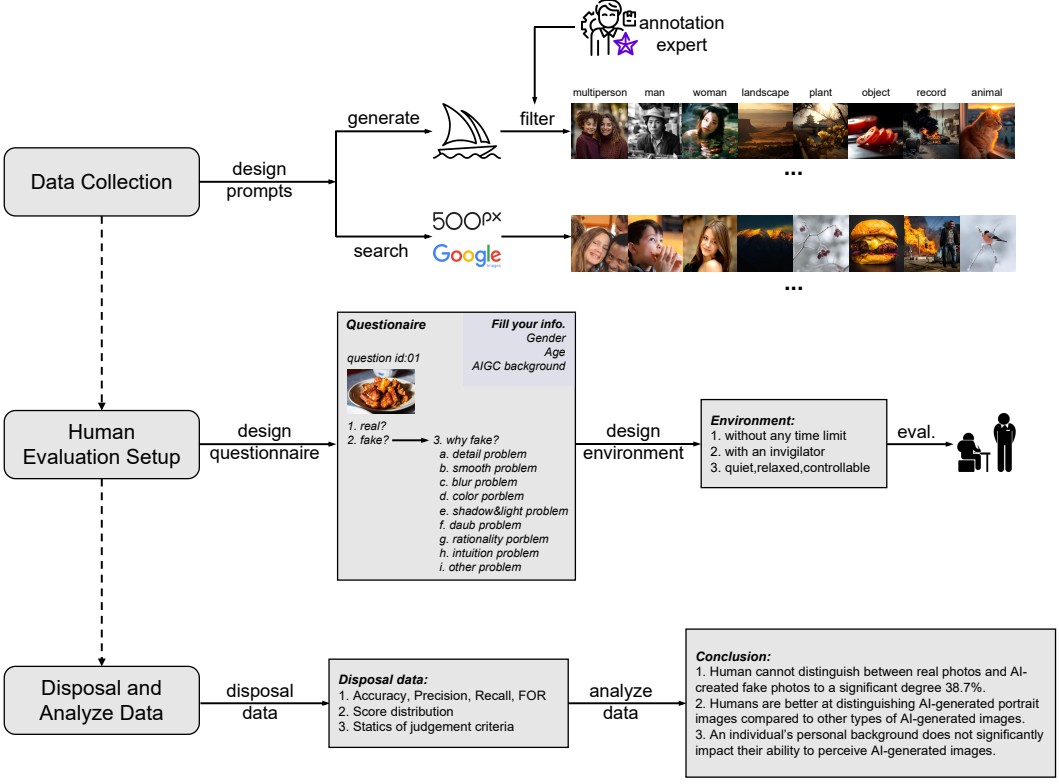

Figure 4: **Procedures for HPBench.**

## C.2 Detail of Human Evaluation

We recruit a total of 50 participants to participate in our human evaluation instead of using crowd-sourcing. In order to ensure comprehensiveness, fairness, and quality of the evaluation, we make efforts to ensure the diversity of the participants. Each participant is asked to complete a questionnaire consisting of 100 questions without any time limit. The questionnaires are completed in the presence of a project team member to guide the participant and ensure the quality of the human evaluation results. It is worth noting that we did not inform the participant about the ratio of real photos to AI-generated images in the questionnaires.

Specifically, each question in the questionnaire provides the participant with an image, and the participant is asked to determine whether the image is generated by AI or not. If the participant thinks that the image is generated by AI, he/she will be required to choose one or more reasons from the eight predefined judgment criteria options or provide their own judgment criteria. The eight options are explained below:

• Detail: AI-generated images may lack fine details, such as wrinkles in clothing or hair details.

• Smooth: AI-generated images may appear smoother or more uniform than real photos, such as smooth skin or unrealistic facial expressions.

• Blur: AI-generated images may be blurry, such as blurry or unclear edges.

• Color: AI-generated images may have unrealistic or inconsistent colors, such as colors that are too bright, too dark, or like the color of animation.

• Shadow & Light: AI-generated images may have unrealistic or inconsistent shadows and lighting, such as shadows or lightning that violate physics.

• Daub: AI-generated images may contain rough, uneven, or poorly applied colors or textures.

• Rationality: AI-generated images may contain irrational/illogical/contradictory contents.

• Intuition: people may judge whether a photo is AI-generated or not by intuition and cannot describe the exact reasons.

It is important to point out that each questionnaire consists of 50 real images and 50 AI-generated images, all of which are randomly sampled from the real image database containing 244 images and the AI-generated image database containing 151 images.

### C.3 Reason of High-quality Fifty-participant Human Evaluation

Inspired by Robert *et al.* [19], we aim to collect high-quality human evaluation data instead of noisy crowdsourcing data to ensure the high quality of the results. Our human evaluation, with a limited number of participants in a controlled environment, and conducting multiple experiments, is commonly referred to as the "small-N design". As the article "Small is beautiful: In defense of the small-N design" [48] suggests, the "small-N design" is the core of high-quality psychophysics. Crowdsourcing involves many participants in an uncontrollable, noisy setting, with each performing fewer trials. In contrast, we strive to ensure that each participant, with diverse backgrounds, takes the full 100-question survey in a quiet, relaxed, controllable, and monitored environment. Those factors contribute to high data quality.

### C.4 Crowd Sourcing Human Evaluation

We collect 1085 crowd-sourced human evaluation questionnaires to make the entire benchmark more comprehensive. We utilize the same experimental setup as the "High-quality Fifty-participant Human Evaluation" before. As the questionnaires are obtained through crowd-sourcing in an uncontrollable and noisy setting, we do not ask the participants to provide justification for each decision. We collect the accuracy of all the questionnaires, and the accuracy is only 49.9%. This highlights that in a fast-paced, noisy, and uncontrolled environment, people are completely unable to distinguish high-quality AI-generated images.

### C.5 Metrics of HPBench

We employ four commonly used evaluation metrics to analyze our results and highlight their respective meanings in the context of our problem. We define positive samples as AI-generated images and negative samples as real images for our problem, and then calculate Accuracy, Precision, Recall, and False Omission Rate (FOR) in the context of our problem.

**Accuracy** is a statistical measure used to evaluate how well a binary classification test correctly identifies or excludes a condition. In our study, accuracy represents the average precision of humans in distinguishing AI-generated images from real images.

**Precision** is the percentage of predicted positive cases that are actually positive. In our study, high precision represents the proportion of AI-generated images out of the total number of images that are predicted as AI-generated ones.

**Recall** is the percentage of true positive cases that are actually predicted as positive. In our study, recall represents the proportion of AI-generated images that are correctly identified as such out of the total number of AI-generated images.

**FOR** is the percentage of false negatives out of all negative cases. In our problem, FOR represents the proportion of real images misidentified as AI-generated images out of the total number of images.

### C.6 Analysis of the AIGC defects

Based on the user data we collected above, we summarize and show nine shortcomings of the current AIGC, as shown in Fig. 5: (1) "Hand problem" refers to situations where fingers overlap or have unreasonable shapes (multi fingers), resulting in images that are not realistic. (2) "Smoothing

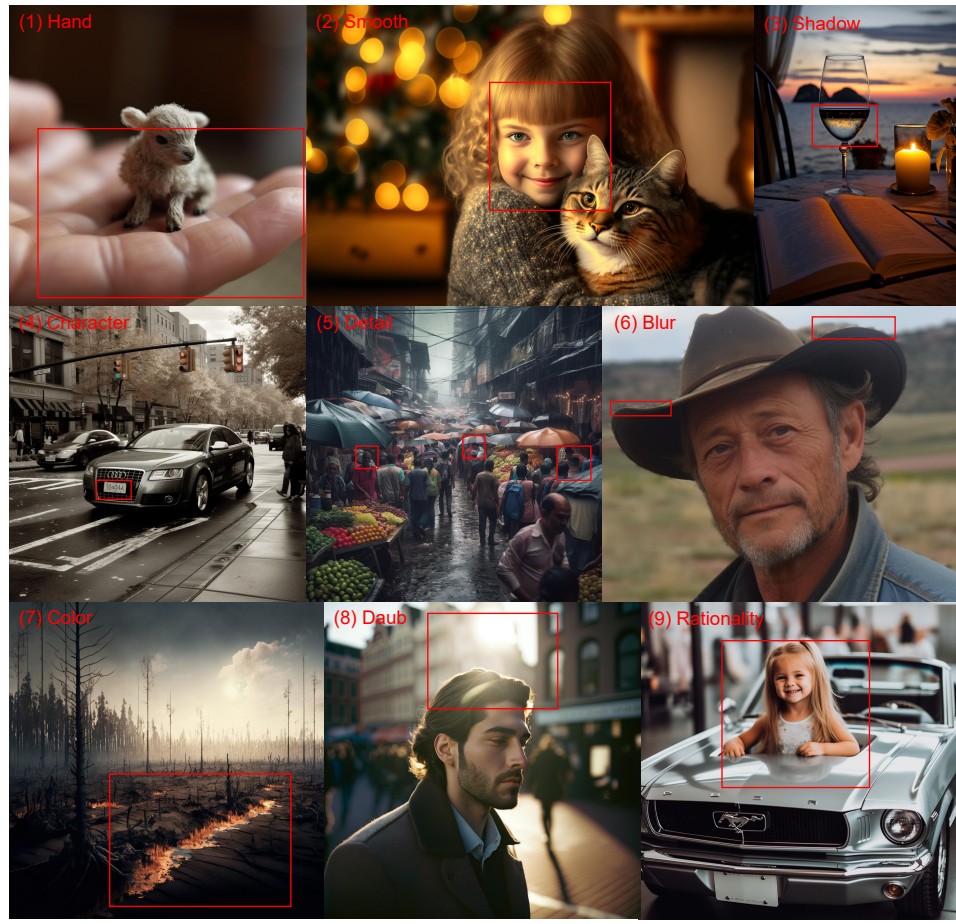

Figure 5: **Nine shortcomings of current AIGC.** We highlight the obvious defects of AIGC with red boxes.

problem" refers to situations where AI-generated images have overly smooth skin, resulting in unrealistic facial expressions and features. (3) "Shadow&Light problem" refers to situations where the position and shape of light sources and shadows in AI-generated images are unreasonable, resulting in unnatural lighting effects. (4) "Character problem" refers to situations where incorrect signs and texts appear in AI-generated images, which do not match reality. (5) "Detail problem" refers to situations where some details in AI-generated images are not realistic or unreasonable, such as wrinkles in clothing or hair details. (6) "Blur problem" refers to situations where AI-generated images are blurry or unclear, resulting in obvious artifacts. (7) "Color problem" refers to situations where the colors in AI-generated images are not realistic or coordinated, such as colors that are too bright, too dark, or like the color of animation. (8) "Daub problem" refers to situations where AI-generated images have been excessively daubed, resulting in lost details or unrealistic images. (9) "Rationality problem" between objects refers to situations where the relationship between objects in AI-generated images is not reasonable, such as incorrect size proportions or unreasonable positions.

### C.7 Detailed Score Distribution

**Detailed score distribution of different categories for all volunteers.** As shown in Fig. 8, we visualize the detailed score distribution of different categories for all volunteers: (a) the detailed score distribution of different categories for all volunteers and all tested images (b) the detailed score distribution of different categories for all volunteers and only AI-generated images (c) the detailed score distribution of different categories for all volunteers and only real images.

**Detailed score distribution of different categories for men and women.** As shown in Fig. 6, we visualize the score distribution of all images for man and woman. We find that the average scores of men and women are almost the same with a relatively accuracy rate of 61%.

We also visualize the detailed score distribution of different categories for man and woman in Fig. 9: (a) the detailed score distribution of different categories for man and woman and all tested images (b) the detailed score distribution of different categories for man and woman and only AI-generated images (c) the detailed score distribution of different categories for man and woman and only real images. A interesting finding is that: Apart from humans having a high recognition rate for human portraits, men have a higher recognition rate for the category *Man* than women, and women have a higher recognition rate for the category *Woman* than men. We speculate that people may have a higher recognition accuracy for more familiar objects.

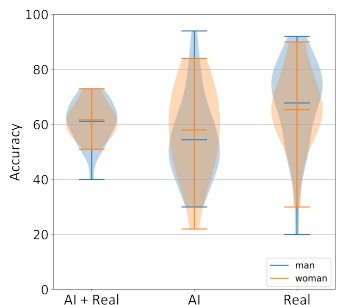

Figure 6: **Human evaluation score distributions calculated by data from men and women.**

**Detailed score distribution of different categories for volunteers with and without AIGC background.** As shown in Fig. 10, we visualize the detailed score distribution of different categories for volunteers with and without AIGC background: (a) the detailed score distribution of different categories for volunteers with and without AIGC background and all tested images (b) the detailed score distribution of different categories for volunteers with and without AIGC background and only AI-generated images (c) the detailed score distribution of different categories for volunteers with and without AIGC backgrounds and only real images.

## D   MPBench

### D.1   More Experiments on MPBench

We conducted more experiments on MPBench, as shown in Tab. 6.

### D.2   Evaluate the best model under the same setting used in HPBench.

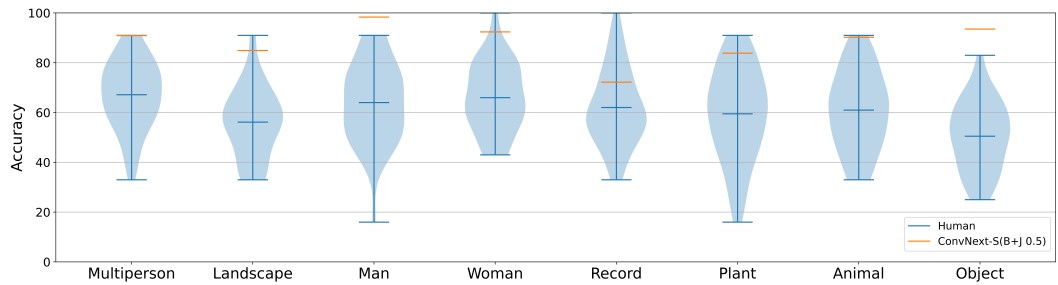

Figure 7: **Human evaluation score distribution and ConvNext-S(B+J 0.5) model score in the same dataset HPBench.**

We also present the results of human and ConvNext-S(B+J 0.5) model on the same dataset HPBench, as shown in Fig. 7. We can find that the results of ConvNext-S are better than human in the results of the two categories: *Man* and *Object*. In the remaining categories, the highest performance of human is better than the performance of the model, but the average performance of human is far worse than the performance of the model.

This demonstrates that it is valuable to study the potential benefits of ensemble the abilities of humans and models in addressing this challenge.

Table 6: **Quantitative comparison of another five models under four training dataset settings with fourteen validation datasets.** "Diff" refers to diffusion model, "AR" refers to autoregressive model and "Unk." refers to unknown model. **Real (R)** denotes the dataset consisting entirely of real images. **Fake (F)** denotes the dataset consisting entirely of fake images. Blue cell denotes the deepfake methods.

| Model | Training Dataset | ImageNet-Test | CelebA-HQ-Train | CC3M-Val | Average Acc. | SD-V2.1-dpm-25 | SD-V1.5-dpm-25 | SD-V1.5Real-dpm-25 | IF-V1.0-dpm++-10 | IF-V1.0-dpm++-25 | IF-V1.0-dpm++-50 | IF-V1.0-ddim-50 | IF-V1.0-ddpm-50 | Cogview2 | Midjourney | StyleGAN3 | Average Acc. | Total Average Acc. |
|---|---|---|---|---|---|---|---|---|---|---|---|---|---|---|---|---|---|---|
| | | | Real | | | | | | | | Fake | | | | | | | R+F |
| | | - | - | - | - | Diff. | Diff. | Diff. | Diff. | Diff. | Diff. | Diff. | Diff. | AR | Unk. | GAN | - | - |
| Swin-S(B+J 0.1) | | 92.4 | 100.0 | 99.9 | 97.4 | 37.9 | 71.7 | 99.9 | 94.4 | 69.6 | 69.4 | 50.6 | 76.9 | 49.5 | 6.8 | 39.5 | 60.5 | 68.4 |
| Swin-S(B+J 0.5) | | 95.1 | 99.9 | 99.9 | 98.3 | 41.3 | 60.2 | 99.9 | 89.7 | 52.3 | 52.2 | 42.0 | 67.6 | 64.5 | 9.5 | 25.1 | 54.9 | 64.2 |
| DeiT-S(B+J 0.1) | Dataset Setting A: | 99.7 | 99.9 | 99.9 | 99.8 | 37.8 | 47.3 | 99.9 | 19.6 | 2.6 | 2.2 | 5.2 | 12.9 | 8.5 | 6.2 | 2.1 | 22.2 | 38.8 |
| DeiT-S(B+J 0.5) | SD-V1.5Real-dpms-25 (1M) CC3M-Train (1M) | 99.4 | 99.9 | 99.8 | 99.7 | 46.2 | 51.5 | 99.9 | 21.5 | 4.0 | 3.1 | 6.6 | 13.9 | 4.7 | 8.6 | 2.8 | 23.8 | 40.1 |
| ResNet50(Fourier) [57] | | 42.3 | 7.2 | 4.3 | 17.9 | 95.9 | 97.4 | 97.3 | 96.9 | 96.0 | 95.4 | 98.6 | 96.0 | 95.6 | 94.9 | 97.0 | 96.4 | 79.6 |
| Xception(Patch) [9] | | 57.1 | 58.7 | 55.7 | 57.1 | 31.0 | 34.9 | 31.4 | 36.5 | 34.7 | 36.9 | 35.5 | 37.6 | 49.7 | 39.5 | 42.7 | 37.3 | 41.5 |
| Swin-S(B+J 0.1) | | 88.1 | 99.9 | 99.9 | 95.9 | 0.4 | 3.0 | 0.1 | 99.6 | 99.8 | 99.7 | 2.1 | 22.0 | 3.3 | 0.1 | 0.7 | 30.0 | 44.1 |
| Swin-S(B+J 0.5) | | 81.5 | 99.9 | 99.9 | 93.7 | 0.9 | 9.9 | 0.3 | 99.6 | 99.9 | 99.8 | 6.7 | 42.6 | 6.8 | 0.1 | 3.3 | 33.6 | 46.5 |
| DeiT-S(B+J 0.1) | Dataset Setting B: | 98.3 | 99.7 | 99.7 | 99.2 | 8.9 | 32.0 | 9.5 | 95.3 | 99.2 | 97.4 | 53.6 | 55.0 | 25.4 | 2.8 | 2.5 | 43.7 | 55.6 |
| DeiT-S(B+J 0.5) | IF-V1.0-dpms++-25 (1M) CC3M-Train (1M) | 97.8 | 99.6 | 99.4 | 98.9 | 11.3 | 36.6 | 10.0 | 95.2 | 99.3 | 97.9 | 57.9 | 61.9 | 27.8 | 3.7 | 2.9 | 45.8 | 57.2 |
| ResNet50(Fourier) [57] | | 42.3 | 45.7 | 51.0 | 46.3 | 60.7 | 61.7 | 73.0 | 59.3 | 72.2 | 70.3 | 29.5 | 69.4 | 40.8 | 60.9 | 70.5 | 60.7 | 57.6 |
| Xception(Patch) [9] | | 54.5 | 17.0 | 29.0 | 33.5 | 43.4 | 44.1 | 49.8 | 23.7 | 19.3 | 20.1 | 63.5 | 35.8 | 49.7 | 57.0 | 58.1 | 42.2 | 40.3 |
| Swin-S(B+J 0.1) | | 99.6 | 99.9 | 99.5 | 99.6 | 2.1 | 3.0 | 1.7 | 1.3 | 2.8 | 2.0 | 6.1 | 5.3 | 27.4 | 1.6 | 99.2 | 13.8 | 32.2 |
| Swin-S(B+J 0.5) | | 99.6 | 99.9 | 99.1 | 99.5 | 1.6 | 2.7 | 1.6 | 2.6 | 4.3 | 3.8 | 4.5 | 3.3 | 23.4 | 1.7 | 99.3 | 13.5 | 31.9 |
| DeiT-S(B+J 0.1) | Dataset Setting C: | 97.7 | 99.8 | 89.9 | 95.8 | 10.5 | 13.4 | 11.2 | 12.4 | 17.8 | 15.4 | 14.9 | 14.5 | 36.6 | 15.9 | 95.9 | 23.5 | 38.9 |
| DeiT-S(B+J 0.5) | StyleGAN3 (87K) | 97.1 | 99.5 | 87.1 | 94.5 | 11.3 | 13.6 | 10.4 | 13.4 | 20.7 | 17.6 | 15.8 | 15.2 | 39.5 | 18.5 | 94.6 | 24.6 | 39.5 |
| ResNet50(Fourier) [57] | StyleGAN3-Train (87K) | 99.2 | 99.9 | 99.2 | 99.4 | 0.1 | 0.2 | 0.1 | 0.6 | 0.2 | 0.1 | 0.2 | 0.5 | 1.0 | 0.6 | 0.45 | 0.3 | 21.5 |
| Xception(Patch) [9] | | 53.3 | 51.8 | 51.2 | 52.1 | 49.7 | 48.7 | 48.5 | 48.6 | 51.2 | 51.6 | 45.1 | 50.8 | 54.7 | 45.9 | 49.0 | 49.4 | 50.0 |
| $F^3$-Net [40] | | 91.7 | 98.8 | 86.2 | 92.2 | 15.4 | 13.3 | 9.5 | 9.7 | 17.6 | 15.3 | 19.9 | 21.7 | 29.4 | 18.1 | 97.5 | 24.3 | 38.8 |
| Gramnet [29] | | 85.1 | 99.5 | 82.2 | 88.9 | 14.3 | 14.6 | 9.5 | 13.1 | 22.0 | 21.0 | 19.2 | 18.4 | 39.9 | 27.1 | 94.3 | 26.6 | 40.0 |
| ELA-Xception [21] | | 73.8 | 99.7 | 68.6 | 80.7 | 49.1 | 38.2 | 35.1 | 41.4 | 42.1 | 41.1 | 47.3 | 53.1 | 73.2 | 38.5 | 89.3 | 49.8 | 56.4 |
| Swin-S(B+J 0.1) | Dataset Setting D: | 83.2 | 99.9 | 99.9 | 94.3 | 48.9 | 92.3 | 99.9 | 99.9 | 99.9 | 99.9 | 67.9 | 95.1 | 61.4 | 13.3 | 99.3 | 79.8 | 82.9 |
| Swin-S(B+J 0.5) | SD-V1.5Real-dpms-25 (460K) | 93.5 | 99.9 | 99.9 | 97.7 | 47.2 | 79.9 | 99.9 | 99.7 | 99.8 | 99.7 | 59.1 | 93.5 | 64.1 | 10.6 | 98.8 | 77.4 | 81.8 |
| DeiT-S(B+J 0.1) | IF-V1.0-dpms++-25 (460K) | 98.2 | 99.9 | 99.6 | 99.2 | 51.0 | 69.4 | 99.8 | 94.6 | 98.4 | 95.0 | 48.0 | 56.5 | 37.3 | 10.3 | 96.7 | 68.6 | 75.3 |
| DeiT-S(B+J 0.5) | StyleGAN3 (87K) CC3M-Train (1M) | 96.7 | 99.7 | 99.0 | 98.4 | 54.8 | 75.7 | 99.8 | 96.0 | 99.1 | 97.6 | 65.7 | 79.1 | 55.4 | 12.1 | 93.1 | 75.3 | 80.2 |
| ResNet50(Fourier) [57] | StyleGAN3-Train (87K) | 58.0 | 26.8 | 62.5 | 49.1 | 41.2 | 51.2 | 48.4 | 43.2 | 50.7 | 53.8 | 7.3 | 53.1 | 21.4 | 58.7 | 64.3 | 44.8 | 45.7 |
| Xception(Patch) [9] | | 75.3 | 69.3 | 69.8 | 71.4 | 51.9 | 54.7 | 50.3 | 55.5 | 59.0 | 58.2 | 32.3 | 55.3 | 47.4 | 43.9 | 17.0 | 47.7 | 52.8 |

## D.3 Hyperparameters of the Experiments

Detailed information about the hyperparameters of the experiments in MPBench are shown in Tab. 7, Tab. 8, Tab. 9, Tab. 10 and Tab. 11.

| config | value |
|---|---|
| optimizer | AdamW |
| optimizer momentum | $\beta_1, \beta_2 = 0.9, 0.999$ |
| weight decay | 0.05 |
| learning rate | 1e-4 |
| learning rate sch. | cosine decay |
| warmup epochs | 0 |
| epochs | 10 |
| augmentation | HFlip, RandomResizedCrop(224), GaussianBlur(0.1), JPEG(0.1) |
| batch size | 1024 |
| dtype | bfloat16 |
| resolution | 224 |
| pretrain | ConvNext-Small-In21k |

(a) ConvNext-S(B+J 0.1)

| config | value |
|---|---|
| optimizer | AdamW |
| optimizer momentum | $\beta_1, \beta_2 = 0.9, 0.999$ |
| weight decay | 0.05 |
| learning rate | 1e-4 |
| learning rate sch. | cosine decay |
| warmup epochs | 0 |
| epochs | 10 |
| augmentation | HFlip, RandomResizedCrop(224), GaussianBlur(0.5), JPEG(0.5) |
| batch size | 1024 |
| dtype | bfloat16 |
| resolution | 224 |
| pretrain | ConvNext-Small-In21k |

(b) ConvNext-S(B+J 0.5)

Table 7: **Settings for ConvNext-S [30] in MPBench.**

# E   Detailed Related Work

## E.1   Image Generation

Generating photorealistic images based on given text descriptions has proven to be a challenging task. Previous GAN-based approaches [7, 20, 27, 56] were only effective within specific domains and datasets, assuming a closed-world setting. However, with the advancements in diffusion models [23, 50], autoregressive transformers [52], and large-scale language encoders [8, 39, 41, 42], significant progress has been made in high-quality photorealistic text-to-image synthesis with arbitrary text descriptions.

State-of-the-art text-to-image synthesis approaches such as DALL·E 2 [43], Imagen [46], Stable Diffusion [44], and Midjourney [4] have demonstrated the possibility of that generating high-quality, photorealistic images with diffusion-based generative models trained on large datasets. Those models have surpassed previous GAN-based models in both fidelity and diversity of generated images, without the instability and mode collapse issues that GANs are prone to. In addition to diffusion models, other autoregressive models such as Make-A-Scene [17], CogView [15], and Parti [55] have also achieved amazing performance. While diffusion models and autoregressive models exhibit impressive image synthesis ability, they all require time-consuming iterative processes to achieve high-quality image sampling. However, the progress made in the field of text-to-image synthesis over the past few years is a testament to the potential of this technology.

## E.2   Deepfake Generation and Detection

In December 2017, a Reddit user going by the pseudonym "Deepfakes" shared pornographic videos created using open-source AI tools capable of swapping faces in images and videos. Since then, the

| config | value |
|---|---|
| optimizer | AdamW |
| optimizer momentum | $\beta_1, \beta_2 = 0.9, 0.999$ |
| weight decay | 0.05 |
| learning rate | 1e-4 |
| learning rate sch. | cosine decay |
| warmup epochs | 0 |
| epochs | 10 |
| augmentation | HFlip, RandomResizedCrop(224), GaussianBlur(0.1), JPEG(0.1) |
| batch size | 1024 |
| dtype | bfloat16 |
| resolution | 224 |
| pretrain | Swin-Small-In1k |

(a) Swin-S(B+J 0.1)

| config | value |
|---|---|
| optimizer | AdamW |
| optimizer momentum | $\beta_1, \beta_2 = 0.9, 0.999$ |
| weight decay | 0.05 |
| learning rate | 1e-4 |
| learning rate sch. | cosine decay |
| warmup epochs | 0 |
| epochs | 10 |
| augmentation | HFlip, RandomResizedCrop(224), GaussianBlur(0.5), JPEG(0.5) |
| batch size | 1024 |
| dtype | bfloat16 |
| resolution | 224 |
| pretrain | Swin-Small-In1k |

(b) Swin-S(B+J 0.5)

Table 8: **Settings for Swin-S [28] in MPBench.**

term "Deepfake" has been widely used to describe the generation of human appearances, particularly facial expressions, through AI methods. The "Malicious Deep Fake Prohibition Act" of 2018 provides a definition of deepfake as videos and audios that have been realistically but falsely altered and are difficult to identify. Similarly, the "DEEP FAKES Accountability Act" of 2019 defines deepfake as media that is capable of authentically depicting an individual who did not actually participate in the production of the content. Yisroel *et al.* [36] defines deepfake as believable media generated by a deep neural network. In essence, deepfake [11] refers to the creation of seemingly realistic but falsified images, audios, videos, and other digital media produced through AI methods, particularly deep learning.

Realistic deepfake media has posed a significant threat to privacy, democracy, national security, and society as a whole. These images and videos have the potential to bypass facial authentication, create political unrest, spread fake news, and even be used for blackmail. The proliferation of fake information through fabricated videos and images can severely undermine our trust in online digital content. Furthermore, the highly realistic nature of deepfake media makes it difficult for humans to identify them as being falsified. Thus, the ability to distinguish between deepfake and real media has become an important, necessary, and urgent matter.

In recent years, there have been many works [6, 9, 13, 16, 34, 37, 38, 54, 57] exploring how to distinguish whether an image is AI-generated. These works focus on images generated by GANs or small generation models [7, 20, 27, 56]. Due to the limited quality of images generated by those methods, it is easy for humans to distinguish whether a photo is AI-generated or not. However, as the quality of generated images continues to improve with the advancement of recent generative models [4, 43, 44, 46], it has become increasingly difficult for humans to identify whether an image is generated by AI or not. Lyu *et al.* [33] provides an in-depth investigation into communication in human-AI co-creation, specifically focusing on the perceptual analysis of paintings generated by

| config | value |
|---|---|
| optimizer | AdamW |
| optimizer momentum | $\beta_1, \beta_2 = 0.9, 0.999$ |
| weight decay | 0.05 |
| learning rate | 1e-4 |
| learning rate sch. | cosine decay |
| warmup epochs | 0 |
| epochs | 10 |
| augmentation | HFlip, RandomResizedCrop(224), GaussianBlur(0.1), JPEG(0.1) |
| batch size | 1024 |
| dtype | bfloat16 |
| resolution | 224 |
| pretrain | DeiT-Small-In1k |

(a) DeiT-S(B+J 0.1)

| config | value |
|---|---|
| optimizer | AdamW |
| optimizer momentum | $\beta_1, \beta_2 = 0.9, 0.999$ |
| weight decay | 0.05 |
| learning rate | 1e-4 |
| learning rate sch. | cosine decay |
| warmup epochs | 0 |
| epochs | 10 |
| augmentation | HFlip, RandomResizedCrop(224), GaussianBlur(0.5), JPEG(0.5) |
| batch size | 1024 |
| dtype | bfloat16 |
| resolution | 224 |
| pretrain | DeiT-Small-In1k |

(b) DeiT-S(B+J 0.5)

Table 9: **Settings for DeiT-S [51] in MPBench.**

a text-to-image system. Instead of exploring the human perception of AI-generated paintings, we study the human perception of AI-generated photographic images that may contain contradictions or absurdities that violate reality. Those AI-generated photorealistic images can potentially pose a significant threat to the accuracy of factual information. In conclusion, the objective of our study is to investigate whether state-of-the-art AI-generated photographic images are capable of deceiving human perception.

## F Discussion, Broader Impact, Limitation and Conclusion

### F.1 Discussion

**Can AIGC deceive humans now?** With the recent rapid advancements in generative AI, AI is now capable of producing highly photorealistic images with rich backgrounds, vivid characters, and beautiful lighting. Although people may able to occasionally differentiate low-quality AI-generated images, it is becoming more and more difficult to distinguish high-quality AI-generated images from real photography. In this study, our human evaluation results indicate that the state-of-the-art (SOTA) AI model is able to deceive the human eye to a significant degree (38.7%). Moreover, our exploration shows that it is no longer reliable to judge whether an image is real based solely on image quality. Instead, people need to consider factors such as over-smoothing portrait faces, coherence, and consistency between objects, and physical laws in the image, which makes the distinguishing process much harder and time-consuming (about 18 seconds for each image in this study).

From another aspect, current AI still can not **consistently** deceive the human eye. AI-generated images still have certain defeats which could be used by humans to distinguish fake images. Besides, creating such high-quality images requires prompt engineering skills and numerous experiments.

| config | value |
| --- | --- |
| optimizer | SGD |
| optimizer momentum | $\beta$=0.9 |
| weight decay | 1e-4 |
| learning rate | 1e-4 |
| learning rate sch. | cosine decay |
| warmup epochs | 0 |
| epochs | 10 |
| augmentation | HFlip(0.5), RandomResizedCrop(224), GaussianBlur(0.1), JPEG(0.1) |
| batch size | 512 |
| dtype | bfloat16 |
| resolution | 224 |
| pretrain | ResNet-50-In1k |

(a) ResNet50(B+J 0.1)

| config | value |
| --- | --- |
| optimizer | SGD |
| optimizer momentum | $\beta$=0.9 |
| weight decay | 1e-4 |
| learning rate | 1e-4 |
| learning rate sch. | cosine decay |
| warmup epochs | 0 |
| epochs | 10 |
| augmentation | HFlip(0.5), RandomResizedCrop(224), GaussianBlur(0.5), JPEG(0.5) |
| batch size | 512 |
| dtype | bfloat16 |
| resolution | 224 |
| pretrain | ResNet-50-In1k |

(b) ResNet50(B+J 0.5)

Table 10: **Settings for ResNet50 [22] in MPBench.**

| config | value |
| --- | --- |
| optimizer | AdamW |
| optimizer momentum | $\beta_1, \beta_2$=0.9, 0.999 |
| weight decay | 0.3 |
| learning rate | 1e-5 |
| learning rate sch. | cosine decay |
| warmup epochs | 0 |
| epochs | 10 |
| augmentation | HFlip(0.5), RandomResizedCrop(224) |
| batch size | 512 |
| dtype | bfloat16 |
| resolution | 224 |
| pretrain | openclip-ViT-L-14 |

Table 11: **Settings for CLIP-ViT-L [10, 41] in MPBench.**

Even though, a few finely adjusted AI images with misleading information can convey wrong ideas and cause enormous damage.

**What the current state-of-the-art image generation model can do and can not do?** Given suitable prompts, the SOTA image generation model can produce photo-realistic images that are indistinguishable from real photographs, as shown in Fig. 1. The prompt can have different formats (e.g., text, image) and arbitrary complexity, including details such as colors, textures, and lighting. There are lots of potential applications for image generation. For instance, AIGC can be used to generate images for advertising campaigns, product catalogs, and fashion magazines. Since it can

easily be controlled by text, AIGC can also be utilized in the film industry to create realistic special effects or even entire scenes, at an extremely low cost. Furthermore, AIGC can be implemented in the gaming industry to produce immersive and lifelike game worlds.

Although generative AI has impressive image generation capabilities, it currently faces several limitations and challenges, as shown in Fig. 5. One of the most significant challenges is generating images of multiple people with intricate details in a single scene. Users can easily infer the authenticity of an image from details. Furthermore, the current model has difficulty generating realistic human hand gestures and positions, which are crucial for many applications such as sign language recognition and virtual reality. In addition, the current state-of-the-art image generation model can produce images with strange details, blurriness, and unrealistic physical phenomena such as lighting issues. These issues limit the model's ability to generate images with high accuracy and fidelity to real-world scenes. Overall, while the SOTA image generation model has shown remarkable capabilities, it still faces significant challenges that need to be addressed for it to achieve even greater success in the field of image generation.

## F.2 Broader Impact

**Societal risks.** As AIGC continues to be promoted in various fields, concerns about its societal use have become increasingly prominent. These concerns involve various issues such as bias and ethics. As we have demonstrated, it is getting more and more difficult for humans to distinguish between AI-generated images and real images. Therefore, AI models may produce content that contradicts or even absurdly violates reality, posing a serious threat to factual information. Photos may then become increasingly difficult to use as evidence in the future, and even serious public opinion effects may result. For example, there were many AI-generated images of Trump being arrested on Twitter recently [2]. Such content may be used to spread false information, incite violence, or harm individuals or organizations. Besides, AIGC can be used to create realistic virtual characters, which may be used for malicious purposes such as online fraud, scams, or harassment.

It is crucial for researchers and practitioners in the field of AIGC to develop strategies to mitigate potential negative impacts. This includes developing methods to identify AI-generated images, establishing guidelines for their ethical use, and raising public awareness about their existence and potential impact. Only by working together can we ensure that the benefits of AIGC are fully realized while minimizing its negative consequences.

**Positive impacts.** Given that AI has shown remarkable performance in creating works of art and photography, it is expected to have a significant impact on artists and photographers in the real world. People can obtain a large number of desired works or photos at a lower cost, which could compress the market for artists and photographers. In this era of fast-food images, where should the new generation of artists and photographers go [1]? However, AI can only generate soulless works, lacking the creativity, imagination, and emotion possessed by human artists and photographers [33]. Even the most advanced AI technology cannot replace the creativity and individuality of human artists and photographers. Therefore, although the emergence of AI has indeed brought new challenges and changes to the fields of art and photography, human artists and photographers are still highly valued.

The emergence of AI technology presents various new opportunities for artists, designers, and users. One of the most significant benefits is the ability to create new and innovative visual works, such as digital art and logos, while reducing the time and cost associated with traditional image creation methods. AI technology allows people to generate unique and novel images that might not have been possible otherwise, leading to new ideas and inspiration. Moreover, AI technology can help optimize existing works of art and photos, leading to improved quality and value. For instance, AI can be used to enhance or restore old or damaged images to their original state [53], which can be particularly useful in restoring historic photographs or artworks [5]. AIGC also provides users a more personalized experience by creating images tailored to their personal preferences [18, 45]. This customization can lead to more engaging and immersive experiences for users.

**Academic impacts.** In this study, we conduct a quantitative human evaluation of whether the most advanced AI model can deceive the human eye. Results indicate several academic directions that could be explored in the future:

- Since people cannot discern the authenticity of images, a natural question arises: *Can AI distinguish whether an image is generated by AI?* Exploring how to use AI to detect AI-generated images is a problem that could be studied [54]. Establishing a detection system to recognize AIGC will greatly ensure the security of society and the credibility of images.

- Even the most advanced image generation model still cannot guarantee the stable generation of high-quality images. At the same time, as shown in Fig. 5, AI-generated images often have certain defects. Our failure case analysis will inspire researchers to design better image generation models. Exploring how to solve these AIGC defects is an important future research direction.

- There is an interesting phenomenon in MPBench: CLIP-ViT-L (LC) [38] freezes the pre-trained backbone and unfreezes the last linear layer. Its generalization in MPBench is very good, but its accuracy in real images has dropped a lot. However, other models initialized from pre-trained models with whole backbone unfrozen have good accuracy in real images, but the generalization in MPBench are not good. This phenomenon shows an interesting research problem: Can we achieve a balance between these two settings? To study how many proportions of backbones should be frozen and how many proportions of backbones should be unfrozen is the best setting for fake image detection task is a good research problem.

- In the real world, it is difficult to obtain comprehensive and diverse data, leading to the famous problem of data imbalance [24]. Using imbalanced data will result in various issues such as the long-tail problem [58] and bias problem [35]. Since the current state-of-the-art image generation model can already produce high-quality data, exploring how to use the image generation model to solve these problems and test the current model's robustness and bias is a problem that could be studied.

## F.3  Limitation

While this work has so far provided several state-of-the-art and large-scale training and validation datasets, as well as several powerful benchmarks, this section explores the limitations of the which are expected to be addressed in future studies.

**Dataset limitation.**   Our training dataset Fake2M only includes three advanced models: Stable Diffusion v1.5 [44], IF [3], and StyleGAN3 [26], limited by the absence of open-source and powerful open vocabulary GAN [25] and Autoregressive models [55]. Due to the lack of API, we are unable to provide a training dataset for Midjourney V5. We hope that future work can further improve the diversity and size of the training dataset to include more powerful generative models.

For the validation datasets, we only include validation datasets for the most advanced generative models, without including validation datasets for other tasks, such as deepfake and low-level tasks. We hope that future work can further improve the diversity of the validation dataset to include more tasks about fake images.

**Benchmark limitation.**   Due to the resource limitations, our high-quality human evaluation HP-Bench only recruits 50 participants. Our human evaluation also lacks diversity in terms of participant background, as it only includes a few attributes such as age, AIGC-background and gender. We hope that future work can further improve the diversity and size of the participants.

## F.4  Conclusion

In this study, we present a comprehensive evaluation of both human discernment and contemporary AI algorithms in detecting fake images. Our findings reveal that humans can be significantly deceived by current cutting-edge image generation models: high-quality AI-generated images can be comparable to real photographs. In contrast, AI fake image detection algorithms demonstrate a superior ability to distinguish authentic images from fakes. Despite this, our research highlights that existing AI algorithms, with a considerable misclassification rate of 13%, still face significant challenges. We anticipate that our proposed dataset, **Fake2M**, and our dual benchmarks, **HPBench** and **MPBench**, will invigorate further research in this area and assist researchers in crafting secure and reliable AI-generated content systems. As we advance in this technological era, it is crucial to prioritize responsible creation and application of generative AI to ensure its benefits are harnessed positively for society.

We have focused on the surprising abilities of the current SOTA image generation model, but we have not addressed the core questions of why and how it achieves such remarkable intelligence, nor the most important issues of how to ensure the security and credibility of AIGC images. It is a significant challenge for researchers to develop secure and reliable AIGC systems that can be trusted for various real-world applications, and ensure the responsible and ethical use of AIGC technology in the future. It is time to prioritize responsible development and the use of generative AI to ensure a positive impact on society.

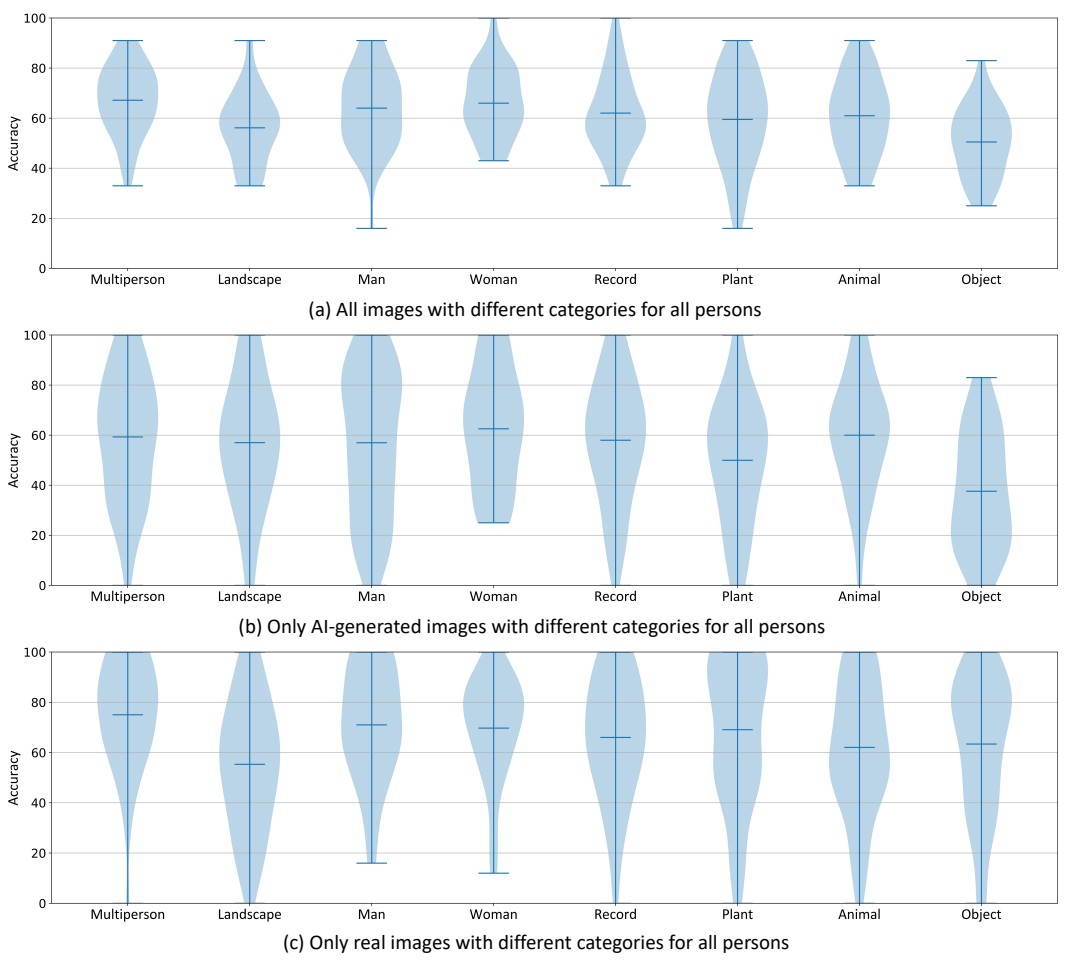

(a) All images with different categories for all persons

(b) Only AI-generated images with different categories for all persons

(c) Only real images with different categories for all persons

Figure 8: **Score distributions for all volunteers with different categories.**

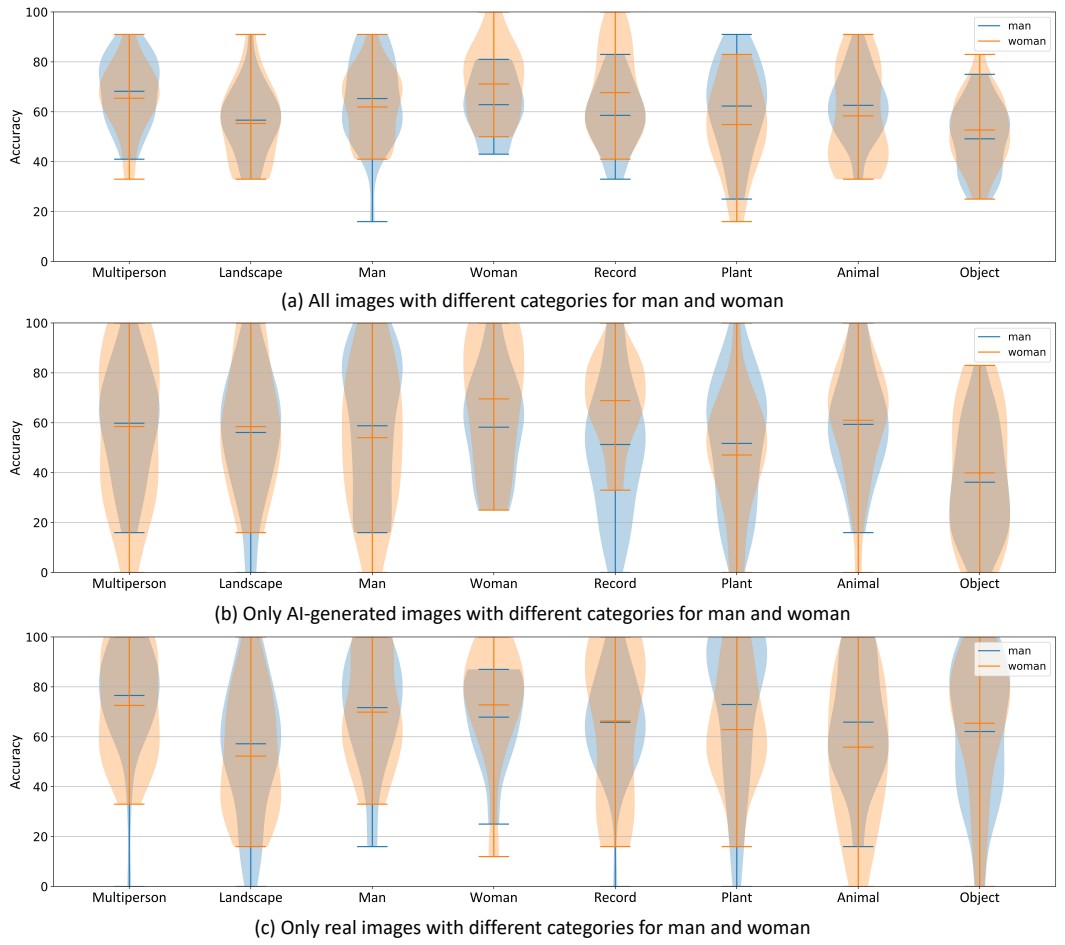

Figure 9: **Score distributions for men and women with different categories.**

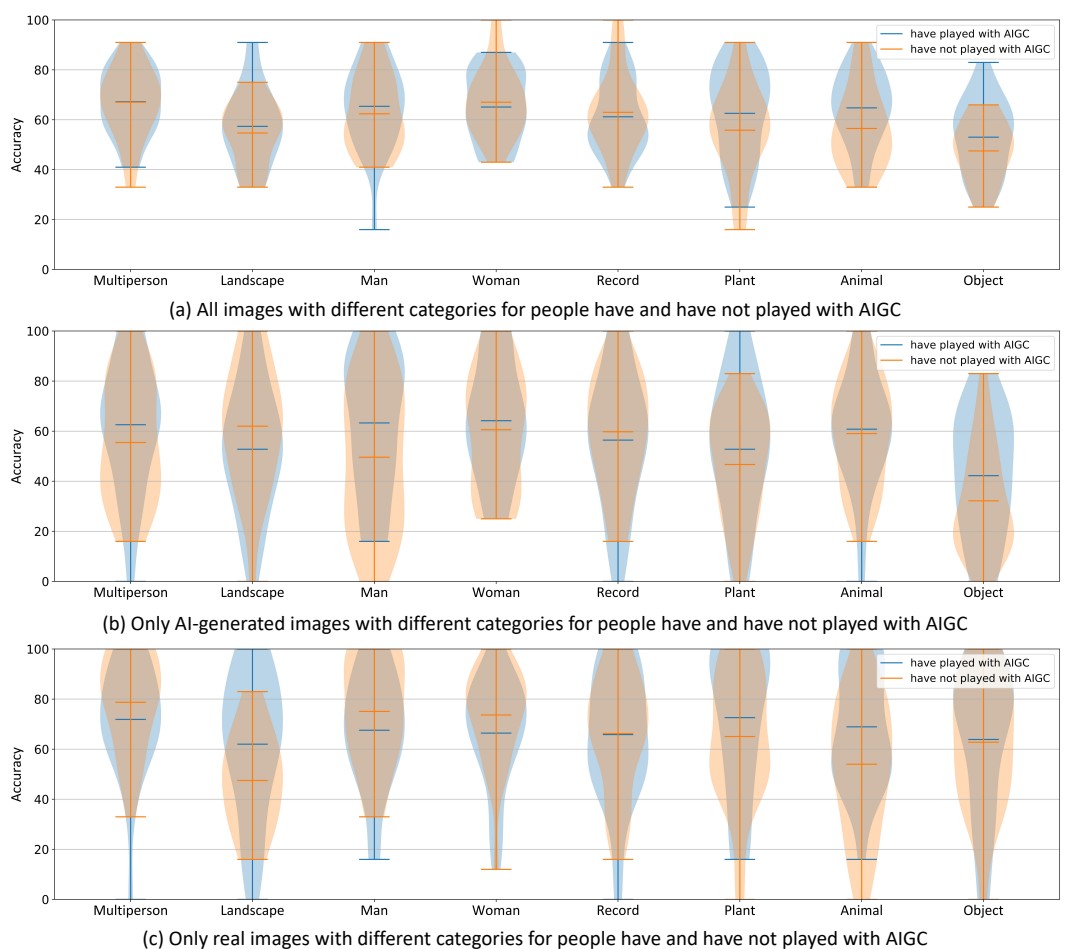

Figure 10: **Score distributions for volunteers with and without AIGC background.**