# OpenReview forum: "Seeing is not always believing: Benchmarking Human and Model Perception of AI-Generated Images"
_NeurIPS.cc/2023/Track/Datasets_and_Benchmarks — NeurIPS 2023 Datasets and Benchmarks Poster_

### Official Review · Reviewer_dyrB · 2023-06-30
**Well motivated with solid contribution**

**Rating:** 6
**Confidence:** 4
**Correctness:** The dataset is constructed in a sound…
**Clarity:** The paper is well-written.

**Strengths:**

•	The paper is well motivated.
•	Creating the dataset is solid contribution.
•	The authors carried out an extensive study focusing on human capability and the proficiency of cutting-edge fake image detection AI algorithms.


**Additional Feedback:**

No additional feedback.

**Documentation:**

The documentation in the provided github link lacks sufficient details on data collection and organization, availability and maintenance.

**Ethics:**

Not applicable.

**Opportunities For Improvement:**

•	The dataset/benchmark website is poorly operated. In fact, there is nothing particular on the GitHub link.
•	The documentation in the provided github link lacks sufficient details on data collection and organization, availability and maintenance.
•	Typo: the sentence in line 67-68 has grammar issue.


**Relation To Prior Work:**

There is no prior work.

**Summary And Contributions:**

This work introduces a new, large-scale dataset, named "Fake2M". This is the largest and most diverse fake image dataset, designed to stimulate and foster advancements in fake image detection research. The authors also establish HPBench, which is a unique benchmark that comprehensively assesses the human capability to discern fake AI-generated images from real ones and MPBench, which is an extensive and comprehensive benchmark including 11 fake validation datasets, designed to rigorously assess the model capability for identifying fake images generated by the most advanced generative models currently available.

---

> ### Author Response · Authors · 2023-08-17
> **Response**
>
> Dear reviewer,
>
> We thank the reviewer for your constructive comments about open source repository construction and your postive feedback! We address your comments below:
>
> > *"The dataset/benchmark website is poorly operated. In fact, there is nothing particular on the GitHub link."*
>
> Thank you very much for your detailed reviewing our [github repository](https://github.com/Inf-imagine/Sentry/tree/main#maintenance-plan)! We are very sorry that we did not update our github link in time. Now the github repository has been completed, the dataset can be found on the huggingface [link](https://huggingface.co/datasets/InfImagine/FakeImageDataset).
>
> > *"The documentation in the provided github link lacks sufficient details on data collection and organization, availability and maintenance."*
>
> Thank you very much for your pertinent suggestion to help us improve the construction of github! Now the github repository has contained [data organization, availability](https://github.com/Inf-imagine/Sentry/tree/main#data-organization) and [maintenance](https://github.com/Inf-imagine/Sentry/tree/main#maintenance-plan) contents. And the data collection can be found in supplementary section B.1.
>
> > *"Typo: the sentence in line 67-68 has grammar issue."*
>
> Thank you very much for your detailed reviewing! We have fixed this mistake.

---

### Official Review · Reviewer_kKjy · 2023-07-13
**The paper relies on pre-existing generative models to generate the synthetic data without introducing novel techniques or approaches. There are concerns about the representativeness of the currently constructed dataset. Since different models, configurations, and sampling strategies can produce a wide range of fake images.**

**Rating:** 5
**Confidence:** 4

**Strengths:**

The need for research on fake image detection has increased as the cases of fake images causing negative impacts on society have been on the rise. Large-scale fake image datasets are expected to promote research in the field of fake image detection and helps people to be less distracted by misinformation or non-existent events.

The key Strengths are:

- a clear description of the methods to collect data
- Interesting findings on human perception depending on the type of photo or the background of the participants
- a large and comprehensive experimental results to establish benchmark (HPBench, MPBench)

**Additional Feedback:**

- listed in weakness section.

**Clarity:**

- The paper is well-organized and easy to follow.

**Correctness:**

- The construction is sound and clear. The primary content and experiments supplied faithfully reflect the claims made by the authors regarding their contributions.

**Documentation:**

- The data and documentation has not yet been released. The github url is currently accessible, but the information is not available. Please make this available when ready.

**Ethics:**

- No issues. I agree with the statement written by the authors.

**Limitations:**

- The technical contribution of this paper appears to be constrained, as the authors solely employ a pre-existing models to produce the synthetic data. I am concerned that the utilization of the current Fake2M dataset will decrease as new generative models emerge in the future. Authors should describe how to maintain and continue to benefit from their datasets and benchmarks.
- It is doubtful that the currently constructed dataset can be representative of the fake samples. Depending on the model or configuration or sampling strategy, a lot of fake images can be produced.
- As it is a large-scale dataset created by a generative model, quality control seems necessary to compose images with good quality.

**Opportunities For Improvement:**

- Outliers can be filtered out through clustering-based method to improve data quality.

**Relation To Prior Work:**

- They show other approaches and describe well.

**Summary And Contributions:**

The paper presents a new collected large-scale fake image dataset (Fake2M) and two benchmarks (HPBench and MPBench)
that assist reliable AI-generated content systems. The dataset consists of 2M training and 257K evaluation data based on the latest generative models.

The key contributions are:

- Large-scale fake dataset for fake image detection research
- A series of baselines and experiments.

---

> ### Author Response · Authors · 2023-08-17
> **Response (1/2)**
>
> Dear reviewer,
>
> We truly thank the reviewer for your careful review and insightful comments. We will address your concerns in turn below:
>
> >*"The technical contribution of this paper appears to be constrained, as the authors solely employ a pre-existing models to produce the synthetic data. I am concerned that the utilization of the current Fake2M dataset will decrease as new generative models emerge in the future."*
>
> We fully understand your concern the utilization of the current Fake2M dataset will decrease as new generative models emerge in the future.
>
> - Firstly, it is important to highlight that, at present, there is still a significant shortage of million-level fake image datasets and a big diverse benchmark. The primary objective behind the creation of Fake2M and MPBench is to provide the research community a large-scale, open-source fake image dataset for research purposes.
>
> - Secondly, since our dataset has covered cross generator setting and cross sampling strategy setting, our dataset and benchmark can provide chance for researchers to find a general and powerful architecture for fake image detection problems and study the generalization problem across both generator and across sampling strategy. The general architecture and generalization methods studied by using our dataset in the future only needs to add the latest fake image data from the latest generation model to our dataset for training. After that they can continuously obtain the most powerful and robust model for general-purpose fake image detection.
>
> It is worth mentioning that Fake2M is just one of the contributions of our work. The core contributions of this paper consist of three parts in main paper lines 74-81. The remaining 2 interesting contributions are as follows:
> - In addition to the dataset Fake2M, we also introduce two new benchmarks, which is an important contribution overlooked by the review. For MPBench, we conduct comprehensive experiments evaluating the abilities of different models under different training settings. Those experiments provide valuable guidance for future fake image model detection design. Besides, we are also the first work benchmarking human ability for fake image detection in HPBench.
>
> > *"Authors should describe how to maintain and continue to benefit from their datasets and benchmarks."*
>
> We very understand your concerns about the maintenance of our datasets. We must admit that maintaining a good benchmark and dataset is a very long-term process. We propose our follow-up plan to support some of the latest models [here]( https://github.com/Inf-imagine/Sentry/tree/main#maintenance-plan). We will regularly update MPBench and our fake image dataset as planned.
>
> **< CONTINUED >**

---

> > ### Author Response · Authors · 2023-08-17
> > **Response (2/2)**
> >
> > > *"It is doubtful that the currently constructed dataset can be representative of the fake samples."*
> >
> > We fully understand your concerns about the representativeness of our dataset. We would like to clarify that our open-source dataset currently represents the optimal choice for researchers studying the Fake Image Detection task. It stands out as the largest and most diverse dataset available, encompassing 11 distinct, up-to-date popular models and 7 different architectures. These architectures cover three different generation paradigms: GAN, Diffusion, and Autoregressive. In Supplementary Table 4, we have provided a detailed comparison of our dataset with other existing datasets. You can also find it [here](https://i.postimg.cc/k5xwGc5Q/2023-08-17-21-06-47.png). To ensure the dataset's representativeness, we have plans to further support additional latest models, continuously expanding the scale and diversity of our fake image dataset, which can be found in [here](https://github.com/Inf-imagine/Sentry#maintenance-plan).
> >
> > > *"Depending on the model or configuration or sampling strategy, a lot of fake images can be produced."*
> >
> > We recognize the reviewer's concern regarding the choice of model and sampling strategy.
> >
> > In fact, it is very difficult to collect fake images of all generation models with all sampling strategy configurations. But our dataset has covered cross generator setting and cross sampling strategy setting. We hope that our dataset and benchmark can provide a platform for researchers to study the generalization problem across generator and across sampling strategy.
> >
> > - For generator, we have collected fake images from 7 different popular modern generators. We hope that our dataset can help researchers solving the generalization problem across the different generators.
> > - For sampling strategy, we tried to change some of the configurations of the sampling process to cover this problem. For example, we change the sampling strategies and sampling steps to generate the following dataset: IF-V1.0-dpm++-10, IF-V1.0-dpm++-25, IF-V1.0-dpm++-50, IF-V1.0-ddim-50 and IF-V1.0-ddpm-50.  We hope that our dataset can help researchers solving the generalization problem of the same generators with different configurations of the sampling process.
> >
> > > *"As it is a large-scale dataset created by a generative model, quality control seems necessary to compose images with good quality."*
> >
> > Thank you for your thoughtful suggestion! We very agree that quality control seems necessary for a large scale dataset created by a generative model. However, for the Fake Image Detection task, low-quality AIGC images also need to be detected as the fake images. A good detector needs to detect high-quality and low-quality fake images at the same time, so we did not drop the low-quality AIGC images. Instead of dropping. we provided the detailed corresponding quality scores using IQA model[1] for each image in [here](https://huggingface.co/datasets/InfImagine/FakeImageDataset). We agree that it is a good research problem to study the influence of quality control for training the fake image detection models. So we hope that our dataset with IQA scores can be used by future researchers to explore the impact of quality control on fake image detection task!
> >
> > [1] Exploring CLIP for Assessing the Look and Feel of Images

---

> ### Author Response · Authors · 2023-08-28
> **Thank you for your time and effort. We are eagerly awaiting your feedback to see if we have addressed your concerns.**
>
> Dear reviewer,
>
> Many thanks for your honest assessments and insightful comments! We have endeavored to supplement and explain our articles based on your thoughful comments. We noticed that there is only ***1 day left until the deadline for discussion***. We want to know if we have addressed your concerns.
>
> Thank you again for your time and valuable suggestions!

---

> > ### Comment · Reviewer_kKjy · 2023-08-28
> >
> > I appreciate your thoughtful response. I have a clearer understanding of the need for Fake2M dataset and plans for maintenance.
> >
> > Nevertheless, one minor concern remains, I hope that the dataset can contribute not only quantitatively but also qualitatively.
> >
> > Since having a large dataset is not enough to train a good image detection model, finding a balance by model, category, and quality seems to be important.
> >
> > Considering measurement method such as IQA, I will re-evaluate the value of the dataset.

---

> > > ### Author Response · Authors · 2023-08-28
> > > **Thank you again for your time, effort and update! We further analyzed the quality issue of our dataset.**
> > >
> > > Dear reviewer,
> > >
> > > Thank you again for the constructive feedback. We are very glad that we addressed your concern about the need and maintenance plan of the Fake2M dataset. We really appreciate your new reply to help us strengthen the paper and will address your new concern in turn below:
> > >
> > > > *"Nevertheless, one minor concern remains, I hope that the dataset can contribute not only quantitatively but also qualitatively. Since having a large dataset is not enough to train a good image detection model, finding a balance by model, category, and quality seems to be important."*
> > >
> > > We fully understand your concern about quality issues with our dataset. After further analysis (can be found in **Supplementary Section B.3 and G**), we believe that our dataset can contribute not only quantitatively but also qualitatively.  We would like to clarify the quality issues, as follows:
> > >
> > > - **Model Balance:** We carefully adjusted the balance of model types during training. We created four training dataset settings to explore the influence of the model type balance problem, as shown in Supplementary Table 6 and Main paper Table 6. We found that using more balanced types of datasets (dataset setting C in Main paper Table 6) for training would get better results in MPBench.
> > > - **Category Balance:** We conducted further data content component analysis to explain that our dataset is a content balanced dataset without data imbalance problem, as shown in Section B.3. To clarify:
> > >     1. There is no content imbalanced problem in Fake2M. We use the captions of famous text-image dataset CC3M to build our Fake2M dataset. In Fake2M, the number of face data is only 82K, accounting for %4 of the total data 2M.
> > >     2. There is no content imbalanced problem in Dataset setting A,B and D. StyleGAN-generated human face images are not ubiquitously used across all our experimental training data settings, as shown in main paper Table 6. As detailed in main paper Table 6, we offer four distinct training-validation settings. Among these, settings A, B, and D employ general fake image datasets.
> > >     3. Dataset setting C is the exception where we've deliberately incorporated human face fake images. This inclusion was intentional, aiming to specifically investigate the performance implications of human face fake images produced by StyleGAN.
> > > - **Quality Balance:** We conducted further analysis of our dataset quality score distirbutions, as shown in Supplementary Section G, Fig.6 and Tab.12. There are three findings, as follows:
> > >     1. The majority of our sub dataset have an average score above 0.6 (with Midjourneyv5-5K having an average score of 0.66)
> > >     2. The average score of all images in the dataset is 0.6.
> > >     3. Only a few datasets (cogview2-22K, IF-ddim-25-15K-1024x1024, IF-ddim-50-15K-1024x1024, stylegan3-r-ffhqu-1024x1024, and stylegan3-r-metfaces-1024x1024) have an average score below 0.6.
> > >
> > >     These demonstrate that our dataset is a high-quality dataset with a large amount of the high-quality images.  The distribution of quality scores across the entire dataset demonstrates a balanced mixture of high-quality and low-quality images, as shown in the "all-images" violin plot of Supplementary Fig. 6. This aligns with our intention: a fake image detection dataset should encompass both high-quality and low-quality image data. In order to better showcase our dataset, we have provided more visualizations about the high quality, mid quality and low quality images in our dataset, as shown in Supplementary Fig.7.
> > >
> > > Therefore, given the analysis of our dataset, we believe that quality issues, as indicated, is effectively mitigated.
> > >
> > > > *"Considering measurement method such as IQA, I will re-evaluate the value of the dataset."*
> > >
> > > We really appreciate your time and efforts to re-evaluate the value of our paper. We are looking forward to your feedback!

---

### Official Review · Reviewer_ZytP · 2023-07-20
**Comprehensive paper**

**Rating:** 7
**Confidence:** 3
**Correctness:** No evident issues with correctness.
**Clarity:** Writing is clear, concise and easy to…

**Strengths:**

- I think MPBench is a valuable benchmark for assessing (1) model's ability to tell apart real/fake images and (2) how good generative models are i.e. they can generate even more realistic images that fool current detection models.
- An interesting finding is that humans struggle significantly to distinguish real photos from AI generated ones, with a misclassification rate of 38.7%. Meanwhile, trained models have better capability to detect real/fake image and the top model achieves a 13% failure rate. This is a good result to raise awareness. I also appreciate the discussion in supp matt section F.

**Additional Feedback:**

The dataset seems promising, and MPBench is a good benchmark. I have some concerns about the credibility of the results from HPBench due to the limited number of participants (and lack of diverse background), and suggest to make the questionnaire publicly available. This paper also lacks comparison to other fake images dataset.

**Documentation:**

There is a link to the training dataset for download.
The 11 validation fake datasets does not seem to be publicly available.
The questionnaire is not publicly available.
As of this review, the training and evaluation code of is not publicly available.

**Limitations:**

My main concern is the credibility of the HPBenchmark.

- The study and analysis on 3.2.1. (ability to distinguish real and AI-generated Images), 3.2.3 (Distinguishability of different photo categories) and 3.2.4 (Results of the Judgment Criteria and Analysis of the AIGC Defect) hinges on 50 participants, and 100 photos (50 fake and 50 real) were used in each questionnaire, which is a rather limited number.
- A possible suggestion would be to make this questionnaire publicly available for (1) reviewers to examine the question asked (2) for anyone to try it out to understand their own literacy to distinguish real/fake image (3) and if users are willing to contribute their data to the questionnaire, it would increase the credibility of the study (3.2.1 to 3.2.4).
- L188- 189 “ participants with AIGC background perform significantly better, with a boost of 3.7%”. Is there any p-value to determine statistical significance?
- In addition, the study on 3.2.2 (ability of participants with various personal backgrounds) comprises 27 participants with AIGC and 23 participants without AIGC background. It seems to be a rather limited data pool to draw such a conclusion from.

**Opportunities For Improvement:**

- There is limited description about the Fake2M dataset in the main paper, whilst majority of the paper focuses on HPBench, MPBench and their findings. There could be more description of the collection pipeline, what prompts are used, what categories of images are collected etc.
- It was claimed as the "largest and most diverse " fake image dataset. However, I don't see any comparison to other datasets in terms of number of images, categories collected in the main paper.

**Relation To Prior Work:**

Is able to distinguish from prior works

**Summary And Contributions:**

1. introduced a large-scale dataset ("Fake2M"), by far the largest and most diverse fake image dataset
2. established HPBench, a unique benchmark that comprehensively assesses the human capability to discern fake AI-generated images from real ones.
3. established MPBench, an extensive and comprehensive benchmark including 11 fake validation datasets, designed to assess the model capability for identifying fake images generated by the most advanced generative models currently available

---

> ### Author Response · Authors · 2023-08-17
> **Response**
>
> Dear reviewer,
>
> We truly thank the reviewer for your constructive comments and your positive feedback! We address each of your concerns in turn below:
>
> > *"There is limited description about the Fake2M dataset in the main paper, whilst majority of the paper focuses on HPBench, MPBench and their findings. There could be more description of the collection pipeline, what prompts are used, what categories of images are collected etc."*
>
> Thank you very much for your constructive comments! We agree that it is essential to provide additional information regarding the themes, generation process/seed, and visualization of "typical" images. To address this, we have included a more detailed description of the collection pipeline, including the prompts used and the categories of images, in Supplementary Table 1, Table 2, and Table 3. Additionally, you can find the visualization of the "typical" images [here](https://github.com/Inf-imagine/Sentry/tree/main/assets/visualization).
>
> > *"It was claimed as the "largest and most diverse " fake image dataset. However, I don't see any comparison to other datasets in terms of number of images, categories collected in the main paper."*
>
> Thank you for your constructive comments regarding the comparison with other datasets. The comparison to other datasets is [here](https://i.postimg.cc/k5xwGc5Q/2023-08-17-21-06-47.png). We have added this in Supplementary Table 4.
>
> > *"The study and analysis on 3.2.1. (ability to distinguish real and AI-generated Images), 3.2.3 (Distinguishability of different photo categories) and 3.2.4 (Results of the Judgment Criteria and Analysis of the AIGC Defect) hinges on 50 participants, and 100 photos (50 fake and 50 real) were used in each questionnaire, which is a rather limited number."*
> >
> >*"In addition, the study on 3.2.2 (ability of participants with various personal backgrounds) comprises 27 participants with AIGC and 23 participants without AIGC background. It seems to be a rather limited data pool to draw such a conclusion from."*
>
> Thank you for your thoughtful comments! We agree that the number of participants is rather limited. So our human evaluation consists of two parts, one is high-quality human evaluation, and the other is crowd sourcing human evaluation. For high quality human evaluation, we only have 50 participants and 100 pictures for the questionnaire. In the Supplementary section C.4, we also provide a crowd sourcing human evaluation of 1085 people using the same questionnaire as the high quality human evaluation. However, we believe that the data from high quality human evaluation is more accurate and reliable compared to crowd sourcing human evaluation. We explained this reason in section C.3 and C.4 of the Supplementary.  Unfortunately, due to questionnaire platform privacy rules, it is not feasible for us to access users' personal information. Therefore, we only have a series of questionnaire results without user information.
>
> >*"A possible suggestion would be to make this questionnaire publicly available for (1) reviewers to examine the question asked (2) for anyone to try it out to understand their own literacy to distinguish real/fake image (3) and if users are willing to contribute their data to the questionnaire, it would increase the credibility of the study (3.2.1 to 3.2.4)."*
>
> Thank you for your imaginative comments about questionnaire! Now we provide a public questionnaire in [here](https://docs.google.com/forms/d/e/1FAIpQLSfhYMAEnqsaxQPNfLqFEpnFxEUqBhmUoRyfPBfYVfZOx4MtLA/viewform?usp=sharing). This questionnaire is currently on Github's repository. Now everyone can participate in our human evaluation to further expand the scale of HPBench!
>
> >*"L188- 189 “ participants with AIGC background perform significantly better, with a boost of 3.7%”. Is there any p-value to determine statistical significance?"*
>
> Thank you for your constructive comments. We found that the p-value of our human evaluation between users with and without AIGC experience is 0.18. This indicates that the result is close to being marginally significant. We have included the p-value result in our paper.

---

> > ### Comment · Reviewer_ZytP · 2023-08-22
> >
> > Thanks for addressing my concerns! I especially appreciate the visualization of the generated images, comparisons to other datasets and the clarification of the p-value. However, the link for the questionnaire is not publicly available.
> >
> > After reviewing the feedback from other reviewers, I find myself in agreement with reviewer gbTx on certain limitations. I'm hesitant to increase my score for now, I'll keep it as is and I look forward to seeing the other reviewer's responses.

---

> > > ### Author Response · Authors · 2023-08-22
> > > **Thank you very much for your reply! We have fixed the questionnaire problem and want to know the limitations that concern you as raised by reviewer gbTx.**
> > >
> > > We really appreciate your reply and your careful review! We are very sorry for the mistake in questionnaire permission setting. Now the [link](https://docs.google.com/forms/d/e/1FAIpQLSfhYMAEnqsaxQPNfLqFEpnFxEUqBhmUoRyfPBfYVfZOx4MtLA/viewform?usp=sharing) is accessible and we have updated it on the [github repository](https://github.com/Inf-imagine/Sentry/tree/main).
> > > At the same time, we want to know the limitations that concern you as raised by reviewer gbTx.
> > > We would like to provide additional explanations and address the concerns expressed by the reviewers.

---

> > > > ### Comment · Reviewer_ZytP · 2023-08-24
> > > >
> > > > Thanks for making the link accessible, the questionnaire was really interesting, and I believe it could be a convincing tool help to garner more public awareness on this subject.
> > > >
> > > > For this question, I share the same concern as reviewer gbTx.
> > > > 1. "Since the MPBench contains face images, some SOTA deepfake detection methods can be evaluated."
> > > >
> > > > Many researchers are only interested in specific categories, and not general fake image detection. The deepfake community is rather large, and I think person or face fake image detection can garner a large amount of interest. Therefore, I think it would be valuable to have categories of object i.e. face, landscapes as a benchmark. In addition, different models i.e. midjourney, stylegan can have very different results for the same category of object (from my personal observation). Therefore, I think it would be good to create subsets for different model types and object categories, and create benchmarks for them.

---

> > > > > ### Author Response · Authors · 2023-08-27
> > > > > **Thank you very much for your reply! We have added some deepfake methods to our MPBench.**
> > > > >
> > > > > We really appreciate your kind reply. We are excited to inform you that we have provided three more methods for deep make detection [1, 2, 3]. We have conducted evaluations of these models not only on the face content dataset StyleGAN3 in MPBench but also on other general content datasets, as shown in Supplementary Table 6.
> > > > >
> > > > > [1] Zhengzhe Liu, Xiaojuan Qi, and Philip H. S. Torr. Global texture enhancement for fake face detection in the wild. IEEE/CVF Conference on Computer Vision and Pattern Recognition, 2020.
> > > > >
> > > > > [2] Yuyang Qian, Guojun Yin, Lu Sheng, Zixuan Chen, and Jing Shao. Thinking in frequency: Face forgery detection by mining frequency-aware clues. European Conference on Computer Vision, 2020.
> > > > >
> > > > > [3] Teddy Surya Gunawan, Siti Amalina Mohammad Hanafiah, Mira Kartiwi, Nanang Ismail, Nor Farahidah Za’bah, and Anis Nurashikin Nordin. Development of photo forensics algorithm by detecting photoshop manipulation using error level analysis. Indonesian Journal of Electrical Engineering and Computer Science, 2017.

---

> > > > > > ### Comment · Reviewer_ZytP · 2023-08-27
> > > > > >
> > > > > > Hi authors,
> > > > > >
> > > > > > I appreciate the evaluation on face content dataset and the addition of extra methods. My concerns have been satisfactorily answered. I have increased the rating, and looking forward to responses from the other reviewers.

---

> > > > > > > ### Author Response · Authors · 2023-08-27
> > > > > > > **Thank you so much for your update!**
> > > > > > >
> > > > > > > We truly appreciate your kind words and very constructive feedback. We are very grateful for your suggestion to make our paper more comprehensive!

---

### Official Review · Reviewer_gbTx · 2023-07-20
**Review of Paper327**

**Rating:** 4
**Confidence:** 5
**Correctness:** yes
**Clarity:** See Opportunities For Improvement and…

**Strengths:**

1.	The target problems of the paper are important and worth exploring.
2.	The paper is easy to follow.


**Additional Feedback:**

N/A

**Documentation:**

yes

**Ethics:**

No concerns

**Limitations:**

This paper does not explicitly discuss the limitations.

**Opportunities For Improvement:**

1.	My main concern for this work is its contribution to developing a good AI-generated Image detection model. Considering that the codes for generating fake images are open-source, it is very easy to generate unlimited data. Thus, how to carefully control and optimize the combination of the images should be a major problem. The misalignment in this work is as follows: (1) The number of images. In Table 2, the number of real images and fake images is different in the training and testing set. The number of fake images from each generator is diverse. (2) The content of images. The content of MPBench includes universal images with diverse object categories and facial images. The human face can be considered as one of the object categories. Adding many face datasets into MPBench makes the proportion of face images become too high. It leads to a content imbalanced problem for training a detection model. The dataset is more like a random combination of existing images and generated images by existing methods. The models trained on this dataset have to consider extra imbalanced data problem.

2.	A severe problem is the limited kind of generators. Most of the generators in this work are basically the same generator in diverse version. For example, F-V1.0-dpms++-10 and F-V1.0-dpms++-25 are the same methods. The existing work [51] shows that training on one generator and testing on another one may lead to performance degradation. The limited kind of generators in this work may restrict the generalization ability. Besides, this paper does not discuss the influence of cross-generator setting.

3.	In Table 6, only two methods[51,40] are compared and discussed. More related works are missed, such as Patch-based classifier[A] and Spec[B]. Some backbones, such as Swin-Transformer and DeiT, can be evaluated. Since the MPBench contains face images, some SOTA deepfake detection methods can be evaluated.

4.	In Table 6, the paper lacks advice for designing detection models. The experiment results cannot inspire people to design good models.


5.	Lacks comparison with the existing datasets. Although the author claims that this is the largest and most diverse fake image dataset, the lack of comparison with the existing dataset makes the statement less convincing.

6.	The models are trained on MPBench and evaluated on HPBench. However, HPBench only contains images from Midjourney. Why not use more generators, such as Stable Diffusion, in constructing HPBench? Only one generator may lead to unfair evaluation.

[A] What makes fake images detectable? understanding properties that generalize. In ECCV, 2020

[B] Detecting and Simulating Artifacts in GAN Fake Images. In WIFS, 2019



**Relation To Prior Work:**

yes

**Summary And Contributions:**

This paper proposes a new benchmark for detecting AI-generated images. Human perception evaluation is performed to show that distinguishing between real and fake images is difficult for human eyes. The existing fake image detection methods are also evaluated in the proposed benchmark.

---

> ### Author Response · Authors · 2023-08-17
> **Response (1/2)**
>
> Dear reviewer,
>
> We truly thank the reviewer for your detailed review and thoughtful comments. We have conducted more experiments that you suggested, revised the paper accordingly, and will address your concerns in turn below:
>
> > *"My main concern for this work is its contribution to developing a good AI-generated Image detection model. ..."*
>
> We appreciate your feedback and would like to clarify the contributions of our work in the following manifolds:
>
> 1. **Dataset Construction**: One of our primary focuses is a new large-scale dataset, named Fake2M, for fake image detection. We agree with the review that generating unlimited data is not difficult (except tons of ‘dirty work’), and carefully controlling and optimizing the combination of the images is the major problem. However, we firmly believe that the difficulty in crafting a dataset shouldn't be the sole determinant of its value. Fake2M, to the best of our knowledge, is the largest and most diverse fake image dataset available, providing an important resource for future fake image detection research and addressing the ever-growing risks associated with generative AI.
> 2. **Benchmarking**: In addition to the dataset, we also introduce two new benchmarks, which is an important contribution overlooked by the review. For MPBench, we conduct comprehensive experiments evaluating the abilities of different models under different training settings. Those experiments provide valuable guidance for future fake image model detection design. Besides, we are also the first work benchmarking human ability for fake image detection.
>
> > *"(1) The number of images. In Table 2, the number of real images and fake images is different in the training and testing set. ..."*
>
> We appreciate the observation made by the reviewer and would like to clarify the design decisions regarding the dataset:
>
> 1. **Training Dataset Considerations**:
>     1. **Balance of Real vs. Fake**: We've maintained a balance between real and fake images during training. Referencing main paper Table 6, Fake2M was utilized in four distinct training settings, with each one having an equal number of real and fake images.
>     2. **Uniformity Across Generators**: We've strived for uniformity in the number of fake images from different generators. For instance, both SD-V1.5Real-ddpms-25 and IF-V1.0-ddpms++-25 were given the same count of 1M images. However, for StyleGAN, we intentionally limited the count to 87K, *to match the training dataset size of stylegan3* (as mentioned in Supplementary Table 3). Generating 1M images from StyleGAN would be impractical, as it could produce redundant or subpar fake images.
> 2. **Testing Dataset Considerations**:
>     1. **Real vs. Fake Consistency**: For the testing set, we ensured near consistency with 139K real images and 252K fake ones.
>     2. **Consistency Across Generators**: Similar to the training dataset, the count of fake images from different generators is kept comparable. Our general target is 15K images for most generators, and a similar number for Cogview2 (22K) and Midjourney (5.5K). For the 60K images of StyleGAN3, they were generated using 6 different generators (as mentioned in Supplementary Table 3), with each generator producing 10K images, which is a similar number for 15K.
>
> We add a more detailed data content component analysis in Supplementary Section B.3.
> We believe that these design decisions contribute to a robust and reliable dataset that serves our research objectives effectively.
>
> > *"(2) The content of images. The content of MPBench includes universal imges with diverse object categories and facial images. ..."*
>
> We recognize the reviewer's concern regarding the potential imbalance stemming from the inclusion of human face images. To clarify:
>
> - There is no content imbalanced problem in Fake2M. In Fake2M, the number of face data is only 82K, accounting for %4 of the total data 2M.
> - There is no content imbalanced problem in Dataset setting A,B and D. StyleGAN-generated human face images are not ubiquitously used across all our experimental training data settings, as shown in main paper Table 6. As detailed in main paper Table 6, we offer four distinct training-validation settings. Among these, settings A, B, and D employ general fake image datasets.
> - Dataset setting C is the exception where we've deliberately incorporated human face fake images. This inclusion was intentional, aiming to specifically investigate the performance implications of human face fake images produced by StyleGAN.
>
> Therefore, given the design and structure of our training sets (as shown in Supplementary Table 5), we believe that the issue of content imbalance, as indicated, is effectively mitigated.
>
> **< CONTINUED >**

---

> > ### Author Response · Authors · 2023-08-17
> > **Response (2/2)**
> >
> > > *"A severe problem is the limited kind of generators. ..."*
> >
> > To the best of our knowledge, Fake2M is the largest and most diverse fake image dataset.
> >
> > **Representativeness**: Our dataset contains 7 different popular modern generation models (SDv2.1, SDv1.5,SDv1.5R,IFv1.0,Cogview2,Midjourney,StyleGAN3). These architectures cover three different generation paradigms: GAN, Diffusion, and Autoregressive. To ensure the dataset's representativeness, we have plans to further support additional latest models, continuously expanding the scale and diversity of our fake image dataset. You can find it [here](https://github.com/Inf-imagine/Sentry/tree/main#maintenance-plan).
> >
> > **Comparison with existing datasets**: We also have provided a detailed comparison of our dataset with other existing datasets in Supplementary Table 4, demonstrating our dataset has the most kinds of generators.
> >
> > > *"Besides, this paper does not discuss the influence of cross-generator setting."*
> >
> > Cross- generator setting is one of the most important setting of our paper. Table 1 of the main paper has already contained settings for cross generators: “Dataset setting A” is only trained on SD-V1.5Real-dpms-2 and tested on all other datasets, “Dataset setting B” is only trained on IF-V1.0-dpms++-25 and tested on all other datasets, “Dataset setting C” is only trained on StyleGAN3 and tested on all other datasets.
> >
> > > *"In Table 6, only two methods[51,40] are compared and discussed. ..."*
> >
> > Thank you for your constructive and thoughtful comments about comparing more related works. Now, we have added these two backbones and five new methods under our four dataset settings in Supplementary Table 6 and you can also find it [here](https://i.postimg.cc/9XvfWYvm/2023-08-27-15-49-48.png).
> >
> > > *"Since the MPBench contains face images, some SOTA deepfake detection methods can be evaluated."*
> >
> > We are aiming for a general-purpose fake image detection model for all types of fake images in the internet, not for a specific types of object.  Based on this goal, we built Fake2M dataset, a general fake image dataset where faces only occupy a very small portion (there are only 70K faces in this dataset, accounting for only %3 of a total of 2M fake images). However, we also added three deepfake methods [1,2,3] under deepfake dataset setting C in Supplementary Table 6 and you can also find it [here](https://i.postimg.cc/9XvfWYvm/2023-08-27-15-49-48.png).
> >
> > [1] Global texture enhancement for fake face detection in the wild.
> >
> > [2] Thinking in frequency: Face forgery detection by mining frequency-aware clues.
> >
> > [3] Development of photo forensics algorithm by detecting photoshop manipulation using error level analysis.
> >
> > > *"In Table 6, the paper lacks advice for designing detection models. ..."*
> >
> > Thank you for your thoughtful comments about insights for future research.
> >
> > We think the advice we can offer are the following:
> > 1. As shown in the Supplementary Figure 3, we statistically analyzed nine shortcomings of the current AIGC model through HPBench. Future researchers can design their models based on these 9 defects.
> > 2. There is an interesting phenomenon in MPBench (as shown in main paper Table 6): CLIP-ViT-L (LC) freezes the pre-trained backbone and unfreezes the last linear layer. Its generalization in MPBench is very good, but its accuracy in real images has dropped a lot. However, other models initialized from pre-trained models with whole backbone unfrozen have good accuracy in real images, but the generalization in MPBench are not good. This phenomenon shows an interesting research problem: Can we achieve a balance between these two settings? To study how many proportions of backbones should be frozen and how many proportions of backbones should be unfrozen is the best setting for fake image detection task is a good research problem.
> >
> > We have added our advice in Supplementary Section F.2 “Academic impacts” part.
> >
> > > *"Lacks comparison with the existing datasets. ..."*
> >
> > We thank for your constructive comments about comparison with other datasets. The comparison to other datasets is [here](https://i.postimg.cc/k5xwGc5Q/2023-08-17-21-06-47.png).
> > We have added this in Supplementary Table 4.
> >
> > > *"The models are trained on MPBench and evaluated on HPBench. ..."*
> >
> > We appreciate your insightful observation. Firstly, the models are trained on four dataset setting in Fake2M and evaludated on MPBench, as shown in main paper Table 6. Secondly, the primary intent behind crafting HPBench was to gauge human capability in distinguishing high-quality images produced by the most potent generative models (as highlighted in the main paper, lines 43-45). As of now, Midjourney stands as the most frequently utilized and widely accepted model for this purpose. We acknowledge the evolving nature of the AI field; thus, as stronger generative models emerge, we commit to updating HPBench to reflect the most recent advancements in human performance evaluation.

---

> ### Author Response · Authors · 2023-08-28
> **Thank you for your time and effort. We are eagerly awaiting your feedback to see if we have addressed your concerns.**
>
> Dear reviewer,
>
> Many thanks for your honest assessments and insightful comments! We have endeavored to supplement and explain our articles based on your thoughful comments. We noticed that there is only ***1 day left until the deadline for discussion***. We want to know if we have addressed your concerns.
>
> Thank you again for your time and valuable suggestions!

---

### Official Review · Reviewer_5zTd · 2023-07-23

**Rating:** 6
**Confidence:** 4
**Clarity:** The paper is well written and clear t…

**Strengths:**

1. The paper provides a comprehensive benchmarking analysis of distinguishing real and AI-generated images, covering various categories across different models.

2. The inclusion of Fake2M as a large dataset with diverse image sets and generation models enhances the resource's potential for future benchmarking purposes.


**Additional Feedback:**

N/A

**Correctness:**

The dataset is constructed in a systematic and sound way. The evaluation methods are overall correct but I would love to see a bit more analysis of the results broken down by themes.

**Documentation:**

Yes.

**Ethics:**

I don't see huge ethical implications but i would flag that a more detailed descriptions of what is being included in the dataset (anything copyright related, NSFW, etc.) and what themes are incorporated in the dataset could greatly benefit potential users and consumers of this dataset.

**Limitations:**

1. The lack of information/analysis on the identities of the human raters could hinder understanding potential biases in identifying AI images. Additionally, the skewed representation of more male than female raters and more younger than older raters should be acknowledged and discussed.

2. The implications of the study's findings, particularly with regard to policy, watermarking, and related areas, are currently limited due to the inability to guarantee 100% accuracy and the associated liability concerns. Addressing these limitations could improve the practical applicability of the research.


**Opportunities For Improvement:**

1. A crucial aspect that could strengthen this benchmark dataset is a clear discussion of the benchmark's underlying goal. The paper would benefit from additional descriptions about Fake2M, such as its themes, generation process/seed, and visualization of "typical" images. Furthermore, identifying the tasks for which Fake2M serves as a suitable or unsuitable proxy would enhance its utility.

2. The paper should explore cases where humans most frequently misidentify real versus AI-generated images and where computer programs struggle as well. Here, I would love to get some intuitions on the images where AI/humans struggle vs. easily tell apart. Investigating the potential benefits of ensemble learning in addressing these challenges would also be valuable.

3. Did the authors present the results on paired results of machine vs human results on the same dataset?

4. More characterizations of what images are in Fake2M? Similar to the breakdown we have for Table 1?

Minor:
5. Could the authors characterize in the difference between users with and without AIGC experience – in particular users with experience (orange distribution in Figure 3) seem to have a smaller variance? Can we quantify this using e.g. a test for equality of variances?

6. Could we investigate the sensitivity to prompt engineering in the generation process of the datasets? Probably on a smaller dataset for a proof-of-concept?



**Relation To Prior Work:**

The paper discussed prior work but could be improved further by comparing the qualitative and quantitative trends observed in this paper to older benchmark efforts (maybe on more specialized datasets since this work seems a larger-scale, comprehensive experiment).

**Summary And Contributions:**

Overall this paper provides a comprehensive benchmarking of distinguishing real and AI-generated images; as well as thorough analysis across categories across these models. The major advantage is that Fake2M comprises a large set of images with different generation models. One key area of improvement is that currently after reading the paper, it is unclear what the *goal* of the benchmark is: the paper could be greatly strengthened if the authors could provide more descriptions about what’s in Fake2M (themes, generation process/seed, visualizing “typical” images), and what tasks would Fake2M be a good/bad proxy for?

---

> ### Author Response · Authors · 2023-08-17
> **Response**
>
> Dear reviewer,
>
> We truly appreciate your thorough review, positive feedback, time and efforts taken to help us strengthen the paper even more! We have conducted experiments that you suggested and will address your concerns in turn below:
>
> > *"A crucial aspect that could strengthen this benchmark dataset is a clear discussion of the benchmark's underlying goal."*
>
> The main goal of our benchmark is to assess the capability of both human and deep learning models to discern between fake AI-generated images and real ones (as mentioned in the main paper, lines 77-81). Considering the participants' average accuracy of only 61.3% on HPBench, we developed Fake2M to stimulate and foster advancements in research on detecting AI-generated fake images.
>
> > *"The paper would benefit from additional descriptions about Fake2M..."*
>
> Thank you very much for your valuable feedback on our article. We completely agree that it is essential for us to provide additional information regarding the themes, generation process/seed, and visualization of "typical" images. To address this, we have included the themes and generation process/seed details in Supplementary Section B. Furthermore, we have also provided a link to the visualization of the "typical" images. You can find the visualizations [here](https://github.com/Inf-imagine/Sentry/tree/main/assets/visualization).
>
> > *"Furthermore, identifying the tasks for which Fake2M serves as a suitable or unsuitable proxy would enhance its utility."*
>
> We agree that it is crucial to identify the tasks for which Fake2M is a suitable or unsuitable proxy. Fake2M, the largest and most diverse fake image dataset, primarily serves as a suitable proxy for fake image detection (as mentioned in the main paper, lines 74-76). By leveraging the diverse image-text pairs within Fake2M, researchers can delve into how generated image content can be effectively incorporated into multimodal models (such as CLIP, BLIP), opening avenues for advancements in areas beyond fake image detection.
>
> > *"The paper should explore cases where humans most frequently misidentify real versus AI-generated images and where computer programs struggle as well. ... "*
> >
> >*"Did the authors present the results on paired results of machine vs human results on the same dataset?"*
>
> Thank you for your constructive and thoughtful suggestion! We very agree that it is valuable to study the potential benefits of ensemble the abilities of humans and models in addressing this challenge. We have further evaluated the performance of our best model, ConvNext-S(B+J 0.5), on the HPBench dataset. The human evaluation score distribution and ConvNext-S(B+J 0.5) model score in the same dataset HPBench can be found in supplementary Figure 5. Upon analyzing the results, we observed that the ConvNext-S model outperforms humans in the categories of Man and Object. However, in the remaining categories, while humans achieve the highest performance, the average performance of humans falls significantly short compared to the model's performance. These findings highlight the notable advantages of the ConvNext-S(B+J 0.5) model over human performance, particularly in the Man and Object categories. The evaluation results underscore the potential for leveraging the model's capabilities and human’s capabilities to enhance the overall performance in fake image detection.
>
> >*"More characterizations of what images are in Fake2M? Similar to the breakdown we have for Table 1?"*
>
> Thank you for your thoughtful suggestion to characterize the Fake2M for reviewers and users! Since we use the captions of cc3m, which is a text-image datasets, to generate Fake2M, it is difficult to show you in the form of main paper Table 1. Nevertheless, we have included some visualizations of examples [here](https://github.com/Inf-imagine/Sentry/tree/main/assets/visualization)!
>
> >*"Could the authors characterize in the difference between users with and without AIGC experience..."*
>
> Yes, quantifying the difference is indeed the best approach. Following the suggestion of reviewer ZytP, we used a p-value to quantify this difference. The p-value obtained from our human evaluation, comparing users with and without AIGC experience, is 0.18. This indicates that the conclusion, there are performance differences  between users with and without AIGC experience, is near-marginal significance.
>
> >*"Could we investigate the sensitivity to prompt engineering in the generation process of the datasets? ..."*
>
> We agree that prompt engineering is sensitive for generation process. On one hand, adding additional prompts may introduce biases. For example, if we add “Film texture, Natural lightning, 4K” prompts, all generated images will lean towards photo style. On the other hand, there might not be a universally effective prompt that works well for every model. Considering these factors, we did not add any prompt when generating MPBench and Fake2M, only use the original prompts from CC3M to generate data.

---

> > ### Comment · Reviewer_5zTd · 2023-08-28
> >
> > First of all, I wanna thank the reviewers for their considerable efforts in addressing concerns from all reviewers. The visualization, as well as the comparison to other datasets, has helped me understand what Fake2M dataset can be used for.
> >
> > However, given that (i) we can always generate an unlimited amount of images from generative models with ever-increasing prompt banks (a sentiment shared by several other reviewers) and (ii) fake image detection is likely not interested in detecting *all classes* of images but only several classes, and I am not sure if this particular dataset could be of utility to that goal, I will keep my original score for this round of discussion.
> >
> > Nonetheless, I enjoyed reading the revisions and still think *this is a good paper that would be above the acceptance threshold* and would love to thank the authors for starting a conversation about this thread of benchmarking work.

---

> > > ### Author Response · Authors · 2023-08-28
> > > **Thank you so much for your update!**
> > >
> > > Many thanks for your positive feedback and constructive comments. We are very grateful for your time and efforts to make our paper more clear and comprehensive!

---

### Author Response · Authors · 2023-08-24
**Global Response to All Reviewers**

Dear reviewers,

We appreciate all reviewers for their thorough review, constructive, time and efforts taken to help us strengthen the paper even more. Here are some ***major modifications***, according to reviewers' comments:

* **Dataset Visualizations (Thanks to Reviewer 5zTd, ZytP):** We have provided a [link](https://github.com/Inf-imagine/Sentry/tree/main/assets/visualization) to the visualization of the "typical" images.

* **Dataset Comparison (Thanks to Reviewer gbTx, ZytP):** The comparison to other datasets is [here](https://i.postimg.cc/k5xwGc5Q/2023-08-17-21-06-47.png). We have added this in Supplementary Table 4.

* **Dataset Content Component Analysis (Thanks to Reviewer gbTx):** The dataset content component analysis can be found in Supplementary B.3 and Table 5.

* **Detailed  Dataset Description (Thanks to Reviewer 5zTd, ZytP)):** The detailed dataset description can be found in Supplementary Section B.

* **Open Questionnaire of HPBench (Thanks to Reviewer ZytP):** We provide a public questionnaire in [here](https://docs.google.com/forms/d/e/1FAIpQLSfhYMAEnqsaxQPNfLqFEpnFxEUqBhmUoRyfPBfYVfZOx4MtLA/viewform?usp=sharing). This questionnaire is currently on [github repository](https://github.com/Inf-imagine/Sentry/tree/main#maintenance-plan). Now everyone can participate in our human evaluation to further expand the scale of HPBench!

* **More Experiments on MPBench  (Thanks to Reviewer gbTx):** We add *five new methods and two new backbones* in MPBench under our four dataset settings in Supplementary Table 6 and you can also find it [here](https://i.postimg.cc/9XvfWYvm/2023-08-27-15-49-48.png).

* **Paired Results of Machine vs Human (Thanks to Reviewer 5zTd):** The human evaluation score distribution and ConvNext-S(B+J 0.5) model score in the same dataset HPBench can be found in Supplementary Figure 5.

* **Detailed Corresponding Quality Scores (Thanks to Reviewer kKjy):**  We provided the detailed corresponding quality scores using IQA model for each image in [here](https://huggingface.co/datasets/InfImagine/FakeImageDataset).

* **Detailed Quality Analysis  (Thanks to Reviewer kKjy):** We conducted further analysis of our dataset quality score distributions, as shown in Supplementary Section G, Fig.6 and Tab.12. We provided more visualizations about the high quality, mid quality and low quality images in our dataset, as shown in Supplementary Fig.7.

* **Github Construction (Thanks to Reviewer dyrB):** The github repository has contained [data organization, availability](https://github.com/Inf-imagine/Sentry/tree/main#data-organization) and [maintenance](https://github.com/Inf-imagine/Sentry/tree/main#maintenance-plan) contents. And the data collection can be found in supplementary section B.1.

More detailed information can be found below.

---

### Author Response · Authors · 2023-08-27
**Thank all the reviewers for your time and efforts! Looking forward to the feedback.**

Dear Reviewers,

We sincerely appreciate your constructive comments, thorough review, time and efforts taken to help us strengthen the paper even more. Based on the initial comments, we have submitted responses to each reviewer separately. We kindly remind all reviewers that there are only ***2 days left until the deadline for discussion***. We are looking forward to your feedback on whether these responses have properly addressed your concerns.

Thank you again for your time and valuable suggestions!

---

### Comment · Area_Chair_dSi5 · 2023-08-29
**Please review the rebuttal**

Dear Reviewers,

The author-reviewer discussion period will end in one day. For those reviewers who have not yet review the authors' rebuttals, please do so as soon as possible and consider whether any score adjustments are necessary.

Thanks,
AC

---

### Author Response · Authors · 2023-08-31
**Thank all the reviewers for your time and efforts! Discussion will end in 7 hours.**

Dear Reviewers,

Many thanks for your time and great efforts in reviewing this paper. We have made significant updates to improve our paper, with more experiments, visualizations, analysis and discussions (can be found in ***Global Response***). With the discussion drawing to a close (***will end in 7 hours***), we are looking forward to your feedback. We will try our best to solve your concerns.

Thank you again for your time and valuable suggestions!

---

### Decision · Program_Chairs · 2023-09-22

**Decision:**

Accept (Poster)

**Comment:**

This article addresses the issue of detecting AI-generated images. Considering the opinions of the reviewers, this work has a certain degree of value. Although, as some reviewers have pointed out, there may be some limitations in the future. However, at this moment, I believe it can be beneficial to the community, so I lean towards accepting this paper.